# ARTIFACTSBENCH: BRIDGING THE VISUAL-INTERACTIVE GAP IN LLM CODE GENERATION EVALUATION

## ABSTRACT

The generative capabilities of Large Language Models (LLMs) are rapidly expanding from static code to dynamic, interactive visual artifacts. This progress is bottlenecked by a critical evaluation gap: established benchmarks focus on algorithmic correctness and largely overlook the visual fidelity and interactive integrity that define modern user experiences. To bridge this gap, we introduce **ArtifactsBench**, a benchmark and automated, multimodal evaluation paradigm for visual code generation. Our framework renders each artifact and captures its dynamic behavior via temporal (three-step) screenshots. This visual evidence, alongside the source code, is then assessed by a Multimodal LLM (MLLM)-as-Judge, which is rigorously guided by a **fine-grained, per-task checklist** to ensure holistic and reproducible scoring. We curate **1,825** diverse tasks and evaluate over 30 leading LLMs. Our automated evaluation achieves **94.4%** ranking consistency with WebDev Arena—a de facto gold standard for human preferences in web development—and up to **90.95%** pairwise agreement with human experts. We open-source ArtifactsBench, including the benchmark, evaluation harness, and baseline results at `https://anonymous.4open.science/r/ArtifactsBench-F7F9`, to provide the community with a scalable and accurate tool to accelerate the development of user-centric generative models.

## 1 INTRODUCTION

Large Language Models (LLMs) are reshaping software creation, extending from conventional code/text to interactive visual artifacts (Jaech et al., 2024; Anthropic, 2025; Guo et al., 2025). These enable responsive web interfaces, data visualizations, and mini-games (Jiang et al., 2024; Lu et al., 2025a). We term an "Artifact" a self-contained, model-generated, executable unit (e.g., web widget, visualization) that integrates code, visuals, and interaction. Despite strong generative capability, rigorous evaluation lags: current tools do not holistically judge visual fidelity and dynamics, forming a bottleneck. WebDev Arena captures human preferences via voting but requires manual evaluation and is difficult to scale (LMSYS Org, 2024). Specifically, the "multimodal instructions" encompass text, images, and interaction logic, while the "interactive visual artifacts" refer to the executable outputs designed and generated based upon these instructions.

Prevailing benchmarks emphasize static correctness (e.g., pass@k in HumanEval (Chen et al., 2021)) or non-visual functionality (e.g., SWE-Bench (Jimenez et al., 2023)); visual code generation is typically judged via screenshot replication (Wüst et al., 2024; Yun et al., 2024; Wu et al., 2024a) or DOM proxies (Xu et al., 2025). None capture the holistic quality of interactive artifacts—layout/aesthetics and dynamic behaviors (responses, state transitions, animations) are undermeasured, so evaluation often falls back to costly, subjective manual inspection or unreliable LLM self-evaluation, lacking scale and objectivity. This gap is critical because high-fidelity, functional interactive visual artifacts are central to user experience in modern applications; their quality directly affects real-world adoption.

We ask a central question: ***How can we automatically, holistically, and reliably evaluate an LLM's ability to transform multimodal instructions—spanning text, images, and interaction logic—into high-quality, executable interactive visual artifacts across diverse prompts?*** Existing efforts pro-

Figure 1: Distribution of ArtifactsBench

Table 1: Dataset statistics of ArtifactsBench.

| Statistics | Number | Statistics | Number |
|---|---|---|---|
| **#Problems** | 1825 | **Length** | |
| **Primary Topics** | | **Question Length** | |
| ○ Game | 413 | - *max length* | 32726 |
| ○ SVG Generation | 123 | - *min length* | 184 |
| ○ Web Application | 441 | - *avg length* | 524.9 |
| ○ Simulation | 75 | **CheckList Length** | |
| ○ Data Science | 122 | - *all max length* | 1111 |
| ○ Management System | 314 | - *all min length* | 95 |
| ○ Multimedia Editing | 118 | - *all avg length* | 522.0 |
| ○ Quick Tools | 179 | - *visual avg length* | 549.1 |
| ○ Others | 40 | - *no-visual avg length* | 494.9 |

vide partial answers. WebBench emphasizes DOM alignment and task automation; WebGen-Bench and FullFront target web comprehension/generation and the development process; and WebDev Arena captures human preferences through head-to-head voting (Xu et al., 2025; Lu et al., 2025b; Sun et al., 2025; LMSYS Org, 2024). Yet none jointly assess visual fidelity and interaction dynamics in a fully automated, scalable, and code-aware manner, nor do they provide reproducible, fine-grained diagnostics that reveal strengths and failure modes. A viable framework must therefore go beyond static code to assess functionality, visual presentation, and interactive behavior captured via staged execution (e.g., three-step screenshots), and provide fine-grained diagnostics to guide progress.

Therefore, we introduce ArtifactsBench, a benchmark for evaluating LLMs on interactive visual artifacts. Our design couples deterministic execution with staged visual evidence and checklist-guided, dual-referee judging, enabling holistic, automated, and reproducible evaluation with strong human alignment. We curate **1,825** tasks via a multi-stage pipeline combining expert sourcing with LLM generation and refinement. As shown in Figure 1 and Table 1, tasks span nine domains and are stratified by complexity for fine-grained capability analysis. Our contributions are threefold:

- **The first large-scale, hierarchical benchmark for interactive visual artifacts.** Artifacts-Bench contains **1,825** queries across **nine** domains (e.g., web, SVG, games, simulations; Figure 1), stratified into **Easy/Medium/Hard** tiers for discriminative evaluation. This design supports fine-grained capability analysis rather than a single static-correctness score.

- **A multimodal, automated evaluation pipeline with checklist-guided MLLM-as-Judge.** We execute artifacts in a sandbox, capture **three sequential screenshots** of interaction, and score against **task-specific 10-dimension checklists** using an MLLM referee (dual-referee: Gemini-2.5-Pro and Qwen2.5-VL-72B) (Zheng et al., 2023; Ge et al., 2023; Zhang et al., 2025). The pipeline measures visual fidelity, interactive correctness, and code quality.

- **Strong human alignment and diagnostic insights.** We evaluate **30+** LLMs, conduct a **280-instance** expert study with pairwise agreement up to **90.95%**, and achieve **94.4%** ranking consistency with WebDev Arena. The analysis reveals systematic failure modes on intensive-interactive tasks and the insight that instruction-tuned generalist models often outperform specialist ones in this multimodal creative setting.

## 2 ARTIFACTSBENCH: A BENCHMARK FOR VISUAL CODE GENERATION

### 2.1 OVERVIEW

**ArtifactsBench** comprises **1,825** executable queries covering **nine** primary topics (Figure 1): "Game Development", "SVG Generation", "Web Applications", "Simulations", "Data Science", "Management Systems", "Multimedia Editing", "Quick Tools", and "Others".

Tasks are stratified into **Easy/Medium/Hard** with a target **30%/40%/30%** split, assigned *post hoc* by aggregated performance of 30+ LLMs to preserve discriminative power. Sub-categories enable finer-grained analysis; prompts appear in Appendix 13.

Table 2: Comparison with prior benchmarks. ArtifactsBench is the first to combine high-granularity (**GR**), strong human consistency (**CHA**), automation (**AF**), and direct visual evaluation (**VE**).

| Benchmark | Data Size | Data Source | Primary Task | GR | CHA | AF | VE |
|---|---|---|---|---|---|---|---|
| Humaneval (Chen et al., 2021) | 164 | Human-Written | Algorithmic Tasks | Low | High | ✓ | × |
| SWE-Bench (Jimenez et al., 2023) | 2,294 | GitHub Issues | Repository-level Bug Fixing | Low | High | ✓ | × |
| WebBench (Xu et al., 2025) | 1,000 | Human-Written | Web Task Automation | Mid | Mid | ✓ | × |
| WebGen-Bench (Lu et al., 2025b) | 101 Instructs | Human & GPT-4 | Web Page Generation | Mid | Mid | ✓ | ✓ |
| WebChoreArena (Miyai et al., 2025) | 532 | Curated Tasks | Web Automation (No UI) | Mid | Mid | ✓ | × |
| FullFront (Sun et al., 2025) | 1,800 QA | Model-Synthesized | Web Comprehension/Generation | Mid | Mid | ✓ | ✓ |
| WebDev Arena (LMSYS Org, 2024) | N/A | User-Prompts | Web Design (Human Vote) | Low | High | × | ✓ |
| **ArtifactsBench (Ours)** | 1,825 | Self-Constructed | **Interactive Visual Artifacts** | High | High | ✓ | ✓ |

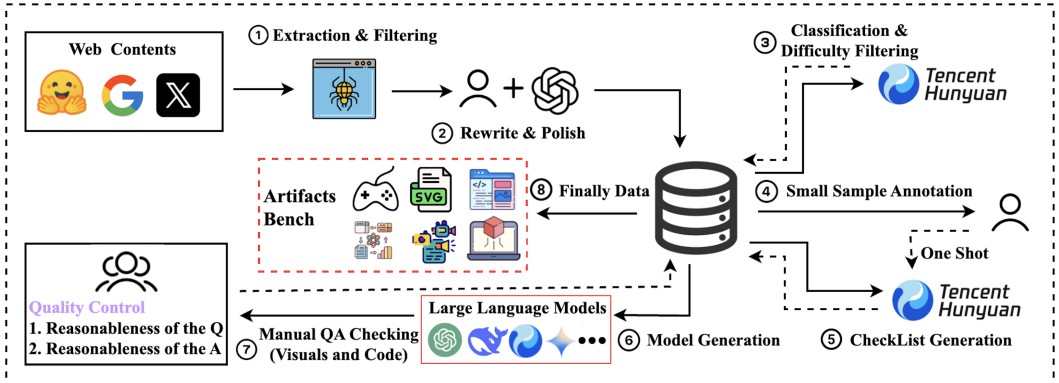

Figure 2: An overview of the data construction process of ArtifactsBench.

## 2.2 DATASET CONSTRUCTION PIPELINE

We aim for ArtifactsBench to be diverse, interactive-first, and rigorously calibrated. Our pipeline (Figure 2) is designed to ensure (i) rich dynamics and real interactivity, (ii) clean and traceable provenance, (iii) strict de-duplication/contamination control, and (iv) reproducibility via standardized execution and checklist-guided judging.

**Sourcing & filtering.** We aggregate candidates from expert showcases, open *SVG/web-snippet* datasets, web case studies, and *LLM visual-to-query* from screenshots. Automated filters drop incomplete, non-visual, license-violating, duplicate, trivial, or non-interactive items, prioritizing dynamics (state changes, animations, responsiveness). The curated pool seeds further expansion.

**De-duplication & contamination control.** Two-stage filtering: (i) MinHash + semantic similarity over prompts, checklists, and normalized DOM/CSS/JS; (ii) screenshot perceptual hashing to catch visually near-identical artifacts. Flagged items are re-authored or discarded.

**Prompt rewriting & difficulty calibration.** Experts rewrite for clarity/completeness/executability; LLM rewriting adds diverse styles; humans verify. Difficulty is assigned *post hoc* by aggregated performance of 30+ models to enforce **30%/40%/30%**; realized split **559/611/655**, stable under leave-family-out resampling. Sub-categories are tagged by a lightweight classifier with human spot-checks (Appendix 13, 14).

**Checklist generation & calibration.** Each query is paired with a **10-dimension** checklist covering visual, interactional, and code qualities. We LLM-draft and then human-refine for specificity and screenability. A **10%** manually curated seed anchors calibration and achieves **Cohen's** $\kappa \geq 0.8$ among annotators; the seed is reused as few-shot references to keep difficulty high and rubrics consistent. The 10 items are grouped into five vision-oriented and five code-oriented dimensions to support diagnostic analysis (Sec. 3).

**Solvability validation & ambiguity repair.** To ensure tasks are answerable and unambiguous, we collect candidate solutions from multiple families and prune underspecified items. Tasks solved only by brittle hacks (e.g., hard-coded coordinates that break under minor viewport changes) are revised or removed. We prefer prompts that admit *multiple acceptable* yet checkable implementations.

Figure 3: The ArtifactsBench evaluation pipeline. The process hinges on a two-stage evaluation: (Step 5) we first validate our MLLM-as-Judge by confirming its high pairwise scoring agreement with human experts on a controlled set of tasks. (Step 6) Once its reliability is established, the automated judge is deployed at scale to evaluate all model outputs across the entire benchmark. The final rankings are then cross-validated against WebDev Arena to ensure alignment with real-world user preferences for visual quality.

**Execution harness & screenshot policy.** We standardize execution with headless Chromium (Playwright) at $1024 \times 768$ resolution and deterministic seeds. We capture **three staged screenshots** (before/during/after scripted interaction) to summarize interaction trajectories so that dynamic feedback and state transitions are evidenced compactly. The same harness is used across models.

**Statistics & coverage.** ArtifactsBench totals **1,825** queries across nine topics (Figure 1; Table 1), with a target Easy/Medium/Hard **30/40/30** split materialized as **559/611/655**. Along the interaction axis, tasks are further divided into Static Visual, Mild-to-Moderate Dynamics, High Dynamics, and Intensive Interactive classes with counts $396/117/536/776$, respectively; classification details and prompts appear in Appendix A.14. Prompts and checklists emphasize *executable* code and observable interaction; purely static items are retained when visual fidelity or structured graphics are the primary goals (e.g., SVG posters).

## 3 EVALUATION METHODOLOGY

To overcome single-metric bias and faithfully capture interaction, we introduce an automated, multi-faceted evaluation framework (Figure 3). The framework targets (i) *score consistency*, defined as stable, reproducible per-query judgments across runs and referees, and (ii) *rank fidelity*, defined as agreement of model orderings with human preferences, evaluated over $Q = 1825$ queries spanning multiple model families. Concretely, we execute each artifact in a standardized sandbox and capture three staged screenshots that summarize the interaction trajectory. We first validate the automatic referee via a 280-instance, six-model human study, demonstrating high pairwise agreement (targeting $>90\%$); detailed annotator protocol and inter-rater statistics are provided in Appendix A.5. After establishing referee reliability, we evaluate all models at scale and verify that the resulting rankings align strongly with WebDev Arena (Figures 7, 8); full settings, cross-judge analysis (including Gemini-2.5-Pro and Qwen2.5-VL-72B), and ablations appear in the appendix.

### 3.1 FINE-GRAINED CHECKLISTS

We use bespoke checklists covering ten dimensions (e.g., functionality, robustness, engineering practices, redundancy, creativity, aesthetics, user experience). Checklists are generated by Hunyuan-Turbos and human-refined; checklist-generation prompts appear in Appendix Figures 15, 16, and final scoring prompts in Figure 17. Each dimension is scored on 10-point scales with detailed rubrics, penalizing redundancy and rewarding innovation; code-level checks (robustness, scalability, performance) reveal issues beyond visual inspection. For balanced assessment and clearer diagnostics, we group ten items into five vision-oriented and five code-oriented dimensions. Concretely, vision-oriented criteria refer to qualities manifesting in rendered appearance and interaction affordances

(layout structure, visual fidelity, motion timing, feedback, UX clarity), while code-oriented criteria evaluate properties visible from code behavior or structure (correctness of logic, robustness to inputs, modularity/engineering hygiene, scalability/performance, and redundancy avoidance).

## 3.2 AUTOMATED EVALUATION AND REFEREES

In contrast to conventional evaluations relying on static code or a single screenshot, our protocol couples interactive evidence with code-aware judging. To make assessment both interactive and reliable, we (i) robustly isolate executable snippets from the model's raw output; (ii) execute in a sandbox and capture three staged screenshots (before/during/after interaction) that summarize the interaction trajectory; and (iii) provide temporal evidence, the original task, the model's full answer, and a fine-grained checklist to the referee model to produce holistic, reproducible scores. The referee aligns visual evidence with task intent and, informed by answer content, audits properties that screenshots alone cannot reveal (e.g., logic correctness, robustness to inputs, modularity/engineering hygiene, and redundancy). This yields consistent judgments across vision- and code-oriented dimensions.

To enhance reproducibility and robustness, we adopt a dual-referee setup: the open-source Qwen2.5-VL-72B and the proprietary Gemini-2.5-Pro. Gemini-2.5-Pro serves as our primary high-capacity referee for main results, while Qwen2.5-VL-72B provides a fully reproducible open-source alternative; we do not fuse their scores, but instead report separate leaderboards and cross-judge partial-order analysis. The two referees induce highly consistent partial-order constraints over model rankings, an empirical outcome of the shared evaluation protocol rather than any enforced calibration between judges, and replacing the deprecated Gemini-2.5-Pro-Preview-0506 preview with the stable Gemini-2.5-Pro leaves our conclusions unchanged. Full settings, cross-judge comparisons, and leaderboards appear in Sec. 4 and the appendix.

## 4 EXPERIMENTS

### 4.1 SETTINGS

We conduct all evaluations in a sandboxed environment with deterministic seeds and fixed rendering settings. Execution uses Playwright (headless Chromium) to render artifacts at a resolution of $1024 \times 768$, capturing three staged screenshots (before, during, and after interaction). Each model is prompted with identical instructions; we apply consistent decoding parameters (temperature and top-p tuned per official recommendations; max tokens sufficient to prevent truncation) and enforce a uniform per-query time budget. Baselines are selected to span (i) multiple families (Qwen2.5/3 (Yang et al., 2024; 2025), DeepSeek (Guo et al., 2025; Liu et al., 2024a), Gemma/GPT/Claude/Gemini (Team et al., 2025; OpenAI, 2023; Anthropic, 2025; Gemini Team & Google, 2023), Seed (Seed et al., 2025), Hunyuan (Liu et al., 2025)), (ii) a range of sizes (from small to flagship), and (iii) modality/training styles (instruction-tuned generalists, coder-specialized, and VL-capable models), ensuring breadth, recency, and reproducibility.

Building on Sec. 3, we evaluate over 30 models using our automated pipeline and dual-referee protocol, analyzing performance across interactivity levels and task categories.

### 4.2 MAIN RESULTS AND ANALYSIS

Proprietary multimodal models (e.g., Gemini-2.5-Pro, Claude 4.0-Sonnet) lead; scores scale with model size and deliberation across families. The largest deficits occur on *Intensive Interactive* tasks and complex *Management System* scenarios, while instruction-tuned generalists consistently outperform specialist coder/VL models. Table 3 and Appendix Figure 9 summarize all baselines scored by our MLLM referee. We report (1) interaction-level metrics across interactivity classes and (2) category metrics across task types, aggregated into one overall score.

**Proprietary multimodal models show clear advantage.** As shown in Table 3, Gemini-2.5-Pro achieves the highest overall score across both open- and closed-source evaluations. Claude 4.0-Sonnet likewise approaches state-of-the-art performance, underscoring the substantial lead that top-tier proprietary multimodal systems currently maintain in this challenging domain.

Table 3: Main results for 30+ LLMs on ArtifactsBench, scored by Gemini-2.5-Pro-Preview-0506 referee. Performance detailed across interactivity levels (**SV**: Static Visual, **MMD**: Mild-to-Moderate Dynamics, **HD**: High Dynamics, **II**: Intensive Interactive) and task categories (**GAME**, **SVG**, **WEB**, **SI**: Simulation, **MS**: Management System). **AVG** is global average. **IFLEN** represents answer length. Since reasoning chain length cannot be obtained for some closed-source models, it is left empty. Proprietary multimodal models lead, and performance scales with model size.

| MODEL | IFLEN | SCORE | | | | | | | | | |
|---|---|---|---|---|---|---|---|---|---|---|---|
| | | SV | MMD | HD | II | GAME | SVG | WEB | SI | MS | AVG |
| *Open-Source Large Language Models* | | | | | | | | | | | |
| Qwen2.5-7B-Instruct | 7905.21 | 29.60 | 26.92 | 29.68 | 24.65 | 24.26 | 23.22 | 29.99 | 23.86 | 28.31 | 27.35 |
| Qwen2.5-14B-Instruct | 6334.34 | 32.07 | 31.93 | 32.83 | 27.85 | 27.73 | 27.89 | 32.77 | 28.11 | 30.57 | 30.49 |
| Qwen2.5-32B-Instruct | 5115.49 | 34.45 | 31.37 | 34.37 | 29.37 | 30.52 | 32.36 | 33.60 | 28.23 | 30.35 | 32.07 |
| Qwen2.5-72B-Instruct | 6029.47 | 35.81 | 35.01 | 36.84 | 32.16 | 33.71 | 34.49 | 36.21 | 29.96 | 33.12 | 34.51 |
| Qwen2.5-VL-72B | 3539.15 | 34.37 | 33.71 | 34.70 | 27.93 | 29.70 | 33.84 | 33.12 | 31.25 | 30.10 | 31.69 |
| Qwen-2.5-Coder7B-Instruct | 5800.23 | 25.58 | 25.80 | 28.80 | 24.34 | 25.21 | 20.58 | 28.56 | 24.49 | 26.27 | 26.01 |
| Qwen-2.5-Coder32B-Instruct | 6318.59 | 37.12 | 36.42 | 37.69 | 32.61 | 32.93 | 33.59 | 37.16 | 34.69 | 33.97 | 35.32 |
| QwQ-32B | 20232.53 | 44.01 | 41.64 | 41.92 | 38.22 | 38.96 | 43.08 | 41.74 | 40.17 | 39.37 | 40.79 |
| Qwen3-4B | 35479.79 | 35.55 | 35.57 | 35.40 | 29.28 | 30.88 | 32.83 | 35.05 | 33.07 | 31.47 | 32.84 |
| Qwen3-8B | 22319.97 | 38.88 | 37.84 | 38.51 | 33.74 | 34.58 | 36.37 | 38.08 | 36.15 | 35.92 | 36.52 |
| Qwen3-14B | 15118.26 | 41.34 | 41.63 | 41.68 | 37.42 | 38.65 | 39.50 | 41.22 | 38.68 | 38.67 | 39.79 |
| Qwen3-32B (Instruct) | 17394.15 | 44.39 | 43.79 | 44.65 | 39.05 | 41.85 | 43.44 | 43.34 | 40.79 | 39.84 | 42.16 |
| Qwen3-30B-A3B (Instruct) | 15772.52 | 42.49 | 40.95 | 42.34 | 37.16 | 39.98 | 42.27 | 41.54 | 38.43 | 37.15 | 40.08 |
| Hunyuan-A13B | 17831.15 | 44.80 | 44.64 | 44.22 | 40.88 | 42.30 | 47.31 | 44.56 | 39.17 | 41.23 | 42.95 |
| Qwen3-253B-A22B (Instruct) | 19400.61 | 47.42 | 46.09 | 46.16 | 41.89 | 44.03 | 47.04 | 45.85 | 43.97 | 42.41 | 44.62 |
| DeepSeek-distill-qwen-32B | 9249.36 | 36.48 | 37.50 | 37.47 | 32.24 | 34.51 | 35.52 | 36.92 | 35.35 | 33.17 | 35.04 |
| Seed-Coder-8B-Instruct | 8934.07 | 36.76 | 37.10 | 37.69 | 32.47 | 34.76 | 36.29 | 36.62 | 33.21 | 32.73 | 35.23 |
| Gemma3-12B-it | 7955.42 | 38.90 | 34.56 | 37.53 | 32.58 | 33.06 | 35.72 | 37.72 | 31.97 | 35.36 | 35.53 |
| Gemma3-27B-it | 7912.14 | 39.97 | 37.63 | 38.80 | 34.54 | 35.62 | 37.18 | 38.65 | 34.49 | 36.01 | 37.16 |
| DeepSeek-V3 | 4518.74 | 38.23 | 37.99 | 37.87 | 32.48 | 34.31 | 37.09 | 37.23 | 34.87 | 33.43 | 35.67 |
| DeepSeek-R1 | 10754.69 | 47.17 | 46.75 | 46.95 | 41.44 | 44.18 | 47.01 | 45.58 | 41.85 | 42.40 | 44.64 |
| DeepSeek-V3-0324 | 11455.42 | 47.78 | 44.43 | 48.53 | 42.55 | 47.58 | 46.34 | 47.47 | 38.71 | 42.88 | 45.56 |
| DeepSeek-R1-0528 | 20780.42 | **51.18** | **53.65** | **51.92** | **51.33** | **51.78** | **52.87** | **50.66** | **50.27** | **45.51** | **51.62** |
| *Closed-Source Large Language Models* | | | | | | | | | | | |
| Seed-thinking-1.5 | 14823.72 | 49.16 | 48.36 | 49.84 | 45.90 | 47.59 | 47.86 | 49.61 | 49.81 | 45.81 | 47.92 |
| GPT-4o | 4883.8926 | 40.60 | 37.74 | 40.32 | 35.04 | 36.96 | 39.54 | 39.27 | 35.73 | 35.83 | 37.97 |
| GPT-4.1-2025-04-14 | 7297.32 | 47.90 | 48.68 | 49.61 | 47.39 | 50.43 | 48.75 | 48.51 | 46.88 | 42.81 | 48.23 |
| O1-2024-12-17 | – | 39.51 | 38.35 | 39.90 | 37.38 | 38.96 | 38.58 | 39.01 | 38.12 | 36.20 | 38.65 |
| OpenAI-o3-mini | – | 46.49 | 45.11 | 46.04 | 43.45 | 45.43 | 46.82 | 45.18 | 43.91 | 41.73 | 44.98 |
| Hunyuan-Turbos-Preview | – | 50.58 | 53.27 | 53.08 | 49.35 | 51.61 | 51.37 | 52.31 | 50.74 | 49.92 | 50.97 |
| Claude 3.7-Sonnet | 15476.18 | 52.73 | 53.54 | 53.48 | 50.83 | 52.24 | 51.63 | 53.64 | 52.14 | 50.27 | 52.19 |
| Claude 4.0-Sonnet | 20633.88 | 57.14 | **59.18** | 57.93 | 53.04 | **57.22** | 56.98 | 55.79 | **56.67** | 53.20 | 55.76 |
| Gemini-2.5-Pro-Preview-0506 | – | 59.02 | 57.69 | 57.99 | 54.70 | 56.65 | **62.37** | 57.28 | 55.26 | 53.04 | 56.79 |
| Gemini-2.5-Pro-Preview-0605 | – | **59.99** | 56.35 | **58.13** | **54.87** | 55.21 | 61.78 | **58.30** | 55.03 | **55.03** | **57.01** |

**Performance scales with model size and deliberation time.** Within the Qwen2.5 and Qwen3 families, performance on ArtifactsBench rises with model capacity. Models with longer inference (sometimes termed "slow thinkers") also score higher, indicating that intricate planning benefits from deeper computation.

**Analysis of open-source model performance.** Among open-source contenders, DeepSeek-R1-0528 sets a new benchmark, indicating that strong code generation and general reasoning aid code-centric visualization. Distillation on limited data yields modest gains: DeepSeek-distill-qwen-32B improves only 3 points over Qwen-2.5-32B yet remains 5 points below Qwen3-32B, consistent with dynamic distillation findings (Liu et al., 2024b).

**Opportunities remain in challenging scenarios.** All models score lowest on Intensive Interactive tasks and complex, project-level visualization (e.g., *Management System*), pointing to clear avenues for improvement.

**Generalist skills outperform specialist expertise.** Instruction-tuned generalist models outperform domain-specific counterparts: Qwen-2.5-Instruct surpasses both Qwen-2.5-coder and Qwen2.5-VL-72B. Producing high-quality visual artifacts requires a synthesis of reasoning, instruction following, and design sense beyond isolated code generation or visual understanding.

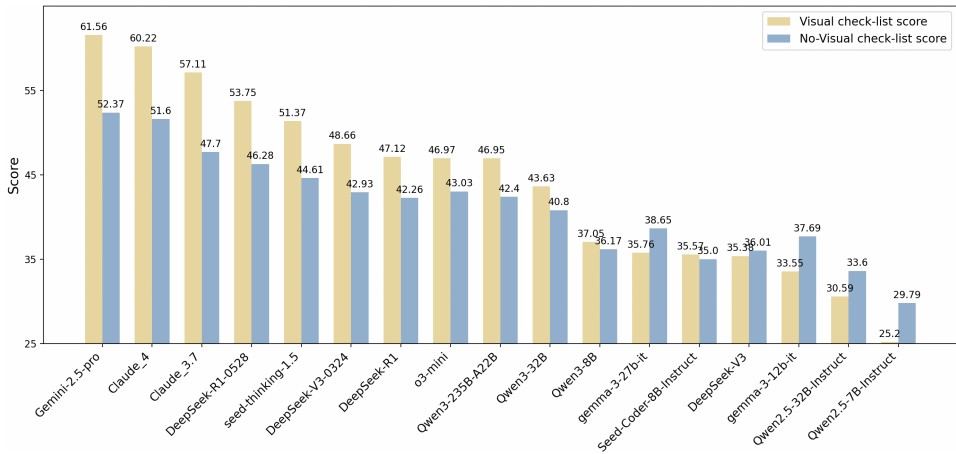

Figure 4: Correlation between vision- and code-oriented scores on ArtifactsBench. The strong positive trend indicates holistic capability and motivates assessing both dimensions.

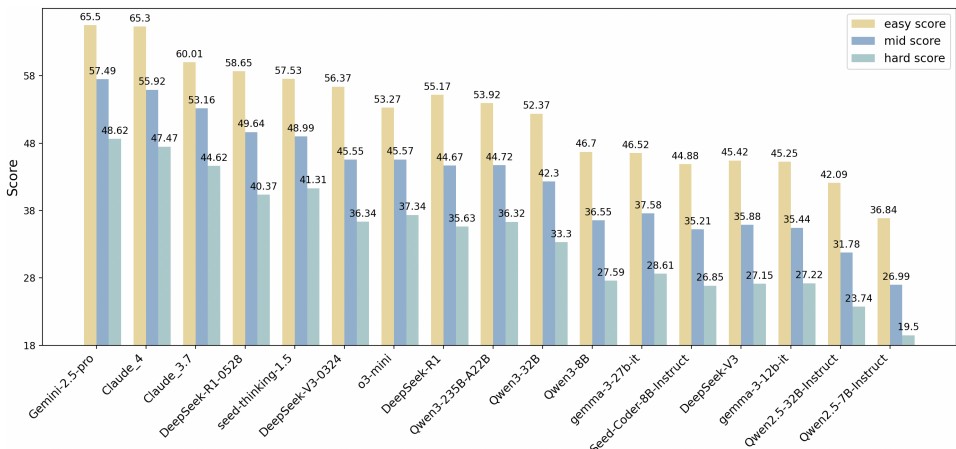

Figure 5: Scores across difficulty tiers on ArtifactsBench.

## 4.3 Fine-grained Analysis

**Vision–Code correlation and difficulty stratification.** We categorize the 10 checklist items into five vision-oriented and five code-oriented criteria. Figure 4 shows a strong positive correlation between vision and code scores, indicating capability improvements are holistic rather than siloed. We further split the benchmark into three tiers; as shown in Figure 5, even the best models struggle on the hardest subset, while relative rankings remain stable across tiers, demonstrating strong discriminative power.

**Human validation and ablations.** We run a double-blind study on 280 queries and 6 models, judged by multiple front-end engineers. Agreement is measured by **Pair ACC**. Incorporating execution screenshots markedly improves agreement, and multiple screenshots better capture dynamics (Table 4). Beyond accuracy, we also study the computational cost of providing visual evidence to the referee: using GPT-4o as a representative model, adding a single execution screenshot increases the average token budget per query by **13.88%**, while three screenshots increase it by **41.51%** (Appendix Table 7). Combined with the gains in Table 4—from **79.06%** pairwise agreement without images to **87.10%** with one image and **90.95%** with three images—this reveals a clear cost–accuracy trade-off: three screenshots capture most of the benefit while keeping budget overhead acceptable. Pushing to five screenshots yields only a marginal improvement to **91.2%** agreement at roughly **20%** additional token cost over the three-screenshot setting, so we adopt three staged screenshots as our default configuration and leave more adaptive screenshot selection to future work. Replacing images with captions helps but remains inferior when visual capability is strong; removing the answer degrades quality, confirming the need for code-aware judging (Table 5).

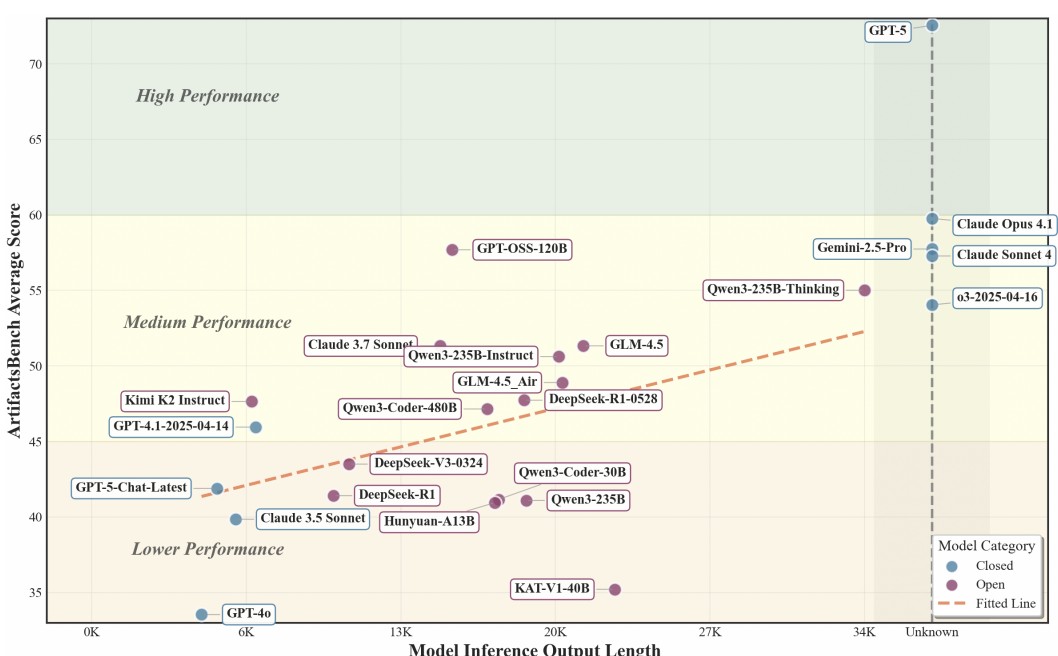

Figure 6: Relationship between model performance and response length on ArtifactsBench (Gemini-2.5-Pro as referee). The trend is broadly positive, but concise, well-structured outputs can remain competitive.

Table 4: Ablation study on multimodal referees with and without images.

| Referee Models w/o img | Pair ACC | Referee Models w/ img | Pair ACC | Referee Models w/ imgs | Pair ACC |
|---|---|---|---|---|---|
| Qwen2.5-VL-72B | 61.80% | Qwen2.5-VL-72B | 68.08% | Qwen2.5-VL-72B | 71.34% |
| GPT-o4-mini | 72.41% | GPT-o4-mini | 76.25% | GPT-o4-mini | 79.41% |
| Gemini-2.5-pro | 79.06% | Gemini-2.5-pro | 87.10% | **Gemini-2.5-pro** | **90.95%** |

Table 5: Ablation study on visual description vs. images and the role of answers.

| Referee Models w/o img | Pair ACC | Referee Models w/ imgs | Pair ACC |
|---|---|---|---|
| Qwen3-32B-Instruct | 64.12 | Qwen3-32B-Instruct w/ caption | 68.78 |
| Qwen2.5-VL-72B | 61.80 | Qwen2.5-VL-72B w/ caption | 66.14 |
| Gemini-2.5-pro | 79.06 | Gemini-2.5-pro w/ caption | 81.74 |
| Gemini-2.5-pro | 79.06 | Gemini-2.5-pro only w/ imgs | 74.94 |
| Gemini-2.5-pro | 79.06 | Gemini-2.5-pro only w/ caption | 70.56 |

In addition, Figure 6 illustrates the relationship between model performance and response length. While longer responses broadly correlate with higher scores (suggesting deeper planning helps), notable efficient models remain competitive with concise outputs—indicating quality over verbosity and the value of compact, well-structured solutions.

**Cross-judge robustness and version stability.** We adopt two SOTA referees with complementary strengths: the open-source Qwen2.5-VL-72B (transparent and reproducible) and the proprietary Gemini-2.5-Pro (higher-capacity reference). They induce highly consistent partial-order constraints over model rankings; residual differences concentrate among top systems, where the stronger referee offers finer resolution. Moreover, replacing the deprecated preview Gemini-2.5-Pro-0506 with the stable Gemini-2.5-Pro yields identical ordering constraints, confirming version stability (see Appendix Figure 11 and Appendix Figure 10).

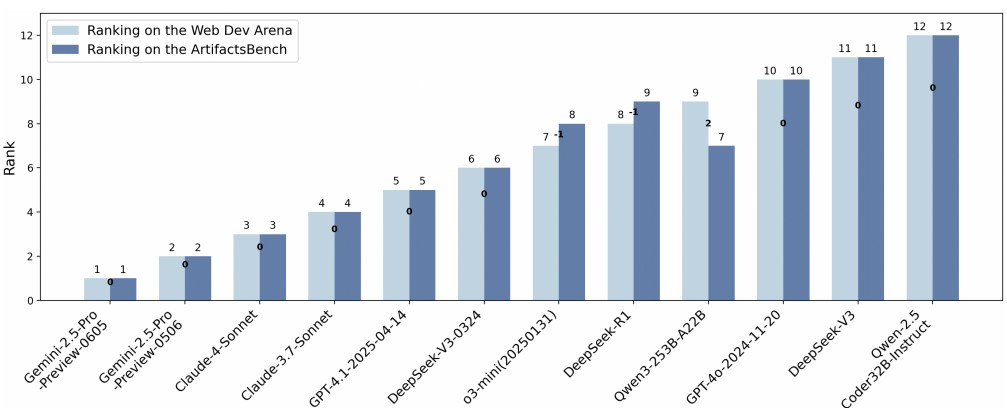

Figure 7: Ranking correlation between ArtifactsBench (judged by Gemini-2.5-pro) and the human-preference-based WebDev Arena. The strong alignment validates that our automated evaluation framework captures qualities that correlate with real-world user perceptions of performance.

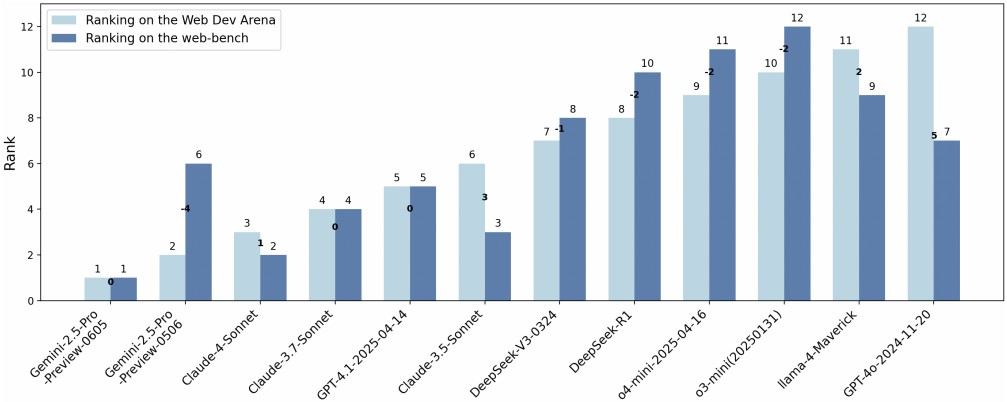

Figure 8: Ranking correlation between WebBench and WebDev Arena. The weaker alignment, compared to ArtifactsBench (Figure 7), suggests that prior static benchmarks may not fully capture the interactive and dynamic qualities prioritized by human users.

### 4.4 ALIGNMENT WITH WEBDEV ARENA

WebDev Arena (Zhou et al., 2023)—a de-facto human-preference gold standard—validates alignment. ArtifactsBench attains **94.4%** normalized Footrule consistency[1] versus **69.4%** for WebBench (Figures 7, 8), indicating stronger reflection of human priorities at scale. Beyond agreement, ArtifactsBench yields actionable diagnostics: for instance, o3-mini ranks slightly higher than human voting because it excels on static code-centric qualities (extensibility, robustness, runtime performance), surfacing strengths that human preference may underweight. We additionally include full leaderboards with both Gemini-2.5-Pro and Qwen2.5-VL-72B referees in the appendix for completeness, while key findings are reported here without deferral.

## 5 RELATED WORK

**Benchmarks for Visual Code Generation.** Foundational benchmarks (HumanEval (Chen et al., 2021), SWE-Bench (Jimenez et al., 2023)) assess algorithms/repositories but not visual fidelity, dynamics, and interaction. Visual-code efforts such as pix2code (Wüst et al., 2024) and Web2Code (Yun et al., 2024) target screenshot-to-code; WebBench (Xu et al., 2025) targets DOM alignment; FullFront (Sun et al., 2025) evaluates the development *process*; WebDev Arena (LMSYS Org, 2024) captures human preference. Design2Code (Si et al., 2025) benchmarks high-fidelity front-end reconstruction from given designs, focusing on static visual fidelity. ArtifactsBench complements these by evaluating live, operable artifacts with complex dynamics via a checklist-driven protocol

---

[1]Based on the $L_1$ distance between rank vectors; closer to 1 is better.

that yields interpretable diagnostics beyond a single score. In contrast to Design2Code's design-to-code setting, ArtifactsBench targets open-ended multimodal instructions and executable, interactive artifacts, emphasizing dynamic behavior, code-level robustness, and interaction-aware evaluation (scripted execution, staged screenshots, and dual MLLM referees) that support fine-grained diagnosis rather than only static reconstruction. It also covers structured graphics (StarVector (Rodriguez et al., 2023), LLM4SVG (Xing et al., 2025)) and dynamic scenes: instead of navigating existing environments (Open CaptchaWorld (Luo et al., 2025)), we assess generating such dynamics via snapshot-based checks (e.g., physics-based mini-games). *Our snapshot-based check is designed to verify that required state transitions and interactive feedback truly occur, and to provide compact, reviewable evidence that enables consistent judging across models and tasks.*

**Evaluation Paradigms for Interactive Visual Artifacts.** DOM/pixel similarity misses high-level semantics and interaction flow. Multimodal LLMs (Gemini Team & Google, 2023; OpenAI, 2023) enable code-aware visual judging within a structured protocol. While these models have shown promising results in perception and QA, they can suffer from hallucination, prompt sensitivity, limited awareness of engineering/code quality, and version drift (Zheng et al., 2023; Ge et al., 2023). Accordingly, ArtifactsBench *adds* structured safeguards—temporal snapshot evidence coupled with fine-grained, task-tailored checklists and a dual-referee protocol (Gemini-2.5-Pro and Qwen2.5-VL-72B)—to improve reliability and interpretability, while keeping this section focused on prior art.

## 6 CONCLUSION

We present **ArtifactsBench**, an automated benchmark for evaluating LLMs on dynamic visual artifacts, addressing static/non-visual gaps. Contributions: (i) a diverse, difficulty-calibrated set of **1,825** tasks; (ii) an automated, checklist-guided MLLM referee scoring code, visuals, and interaction. Across **30+** LLMs, scores align strongly with expert judgments and Web-Dev Arena, offering diagnostics on strengths/failure modes and advancing artifact generation.

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

## A APPENDIX

### A.1 LIMITATIONS AND FUTURE WORK

While ArtifactsBench represents a significant step forward in the automated evaluation of LLM-generated visual code, we recognize its limitations, which in turn open up exciting avenues for future research.

**Deepening the Evaluation of Interactivity.** Our current methodology evaluates dynamic behavior by capturing and analyzing a sequence of screenshots at fixed intervals. This approach effectively assesses many forms of interactivity. However, for highly complex, long-horizon, or state-dependent interactions (e.g., multi-step user workflows in a web application, or the nuanced physics in a game), this discrete sampling may not fully capture the fluidity, correctness, and robustness of the entire interactive experience. Future work could explore more sophisticated dynamic analysis techniques, such as programmatically interacting with the Document Object Model (DOM) to verify state transitions or employing video-based analysis to evaluate the entire user session, thereby enabling a more profound understanding of complex interaction logic.

**Exploring Agentic and Iterative Development.** ArtifactsBench currently focuses on evaluating the quality of the final artifact generated in a single turn from a given prompt. This scope does not assess an LLM's capability to function as an autonomous **agent** that can iteratively refine an artifact based on feedback, debug its own code in response to errors, or plan and execute a multi-step development process. These agentic capabilities are crucial for tackling real-world software engineering challenges. A promising direction for future research is to extend ArtifactsBench into an agentic evaluation framework. In such a setup, the model would need to engage in a multi-turn dialogue with a simulated environment (e.g., a user, a linter, or a debugger) to incrementally build, test, and enhance the visual artifact. This would provide a more realistic testbed for evaluating the end-to-end problem-solving abilities required for truly intelligent and collaborative code generation.

### A.2 ETHICS, FAIRNESS, LICENSING, AND SOCIETAL IMPACT

**Licensing and provenance.** All data sources are curated with clear provenance and licensing. Items with unclear or incompatible licenses are excluded. We apply de-duplication and contamination control over prompts, checklists, code/DOM/CSS/JS, and screenshots (perceptual hashing), and remove any flagged items. The benchmark contains no personally identifiable information (PII).

**Fairness and bias mitigation.** We strive for fair and unbiased evaluation by (i) balancing coverage across nine domains and three difficulty tiers; (ii) adopting *dual referees* (Gemini-2.5-Pro and Qwen2.5-VL-72B) to diversify inductive biases; (iii) using task-specific, fine-grained checklists that anchor objective criteria across vision- and code-oriented dimensions; and (iv) validating with expert studies and alignment to WebDev Arena. Evaluation order and execution settings are standardized to reduce position and presentation bias.

**Sustainability and reproducibility.** We standardize execution (fixed seeds, headless Chromium, staged screenshots) and release scripts to enable efficient replication. To lower cost and environmental impact, we support stratified sub-benchmark evaluation and caching of intermediate artifacts (code extraction, screenshots, and referee inputs), enabling incremental and energy-conscious benchmarking. For the camera-ready version, we have globally reviewed all tables and figures to standardize caption placement (*table captions above, figure captions below*) and corrected previous inconsistencies.

**Safety and privacy.**    All artifacts are executed within a sandbox with strict timeouts and resource limits. External network/file-system access is blocked beyond the sandbox scope. Inputs are sanitized, and no user data is collected.

**Residual limitations.**    Residual biases may persist due to evolving model ecosystems and referee versions; license and contamination checks, while multi-stage, are not perfect; fixed viewport settings may underrepresent some responsive designs. We disclose these limitations and welcome community audits and extensions.

## A.3    DETAILED INTRODUCTION TO DATA COLLECTION AND CLEANING

Our data construction process is a multi-stage pipeline designed to ensure the resulting benchmark is diverse, interaction-focused, and rigorously calibrated. The pipeline emphasizes rich dynamics, clean provenance, strict deduplication, and reproducible, checklist-guided evaluation. The key stages are as follows:

**Sourcing & Filtering:**    We began by aggregating candidate tasks from a wide array of sources, including expert showcases and course experiments, engineering blogs, and front-end code snippet repositories like CodePen and Observable. We also mined open-source SVG and visualization collections, web case studies, and simple game websites. An additional source involved using "LLM visual-to-query" techniques on screenshots to generate task descriptions. This initial pool was then subjected to both automated and manual filtering to discard tasks that were incomplete, non-visual, contained private information, lacked sufficient interactivity, or were overly trivial. Furthermore, we manually curated a significant number of high-interest cases that are popular in user evaluations, such as physics simulations (e.g., bouncing balls in a hexagon) and astronomical models (e.g., a rotating solar system).

**Prompt Rewriting and Polishing:**    Initial prompts were rewritten and enhanced using large language models to generate stylistic variations and add detailed constraints. These augmented prompts then underwent a manual review process to improve their clarity, diversity, and executability.

**Deduplication and Contamination Control:**    To ensure novelty and prevent data leakage, we implemented a rigorous deduplication process. We applied MinHash and semantic similarity checks on task prompts and checklists, and used perceptual hashing for screenshot features to remove near-duplicates and suspect samples. Furthermore, to control for data contamination, we employed a multi-model scoring approach. We identified and filtered out tasks that received universally high scores across these models, as this indicates a high likelihood that they are common examples already present in public training corpora.

**Difficulty and Domain Annotation:**    Tasks were categorized into nine distinct domains. To assign a difficulty level (Easy, Medium, or Hard), we used a combination of inference scores from multiple models fed into a simple classifier, followed by manual sampling and review. This process was designed to achieve a balanced distribution of difficulty tiers across the different domains.

**Checklist Generation and Calibration:**    For evaluation, we generated a 10-dimension checklist for each task. An LLM first drafted the checklist based on a unified template. A human expert then meticulously refined the checklists for a seed set of approximately 10% of the tasks to achieve a high inter-annotator agreement (Cohen's $\kappa \geq 0.8$). This refined seed set was subsequently used as a few-shot reference to ensure stylistic consistency and calibrated difficulty for the LLM-generated checklists for the remaining tasks.

**Solvability Verification and Ambiguity Resolution:** To confirm that tasks were solvable and well-defined, we collected candidate solutions from multiple families of models for every task. This step allowed us to identify and eliminate tasks that were difficult to verify or relied heavily on fragile hacks. Prompts that were underspecified or led to ambiguous but valid solutions were rewritten or removed to ensure every task was both solvable and adjudicable.

### A.4 Detailed Analysis of the Results

**Proprietary multimodal models demonstrate a clear advantage.** As shown in Table 3, Gemini-2.5-Pro achieves the highest overall score across both our open-source and proprietary human evaluations. Claude 4.0-Sonnet likewise approaches state-of-the-art performance, underscoring the substantial lead that top-tier proprietary multimodal systems currently maintain in this challenging domain.

**Performance scales with model size and deliberation time.** Within the Qwen2.5 and Qwen3 model families, we observe a consistent trend: performance on **ArtifactsBench** scales positively with model capacity. Moreover, models engaging in extended inference—so-called "slow-thinkers"—tend to score higher, indicating that the intricate planning required for visual code generation benefits appreciably from deeper computational reasoning.

**Analysis of open-source model performance.** Among the open-source contenders, DeepSeek-R1-0528 sets a new benchmark, demonstrating that models with robust code generation and general reasoning capabilities can excel in code-centric visualization tasks. We also note an important knowledge-distillation finding: DeepSeek-R1-Distill-Qwen-32B improves by only 3 percentage points over its base Qwen-2.5-32B, yet remains 5 points below Qwen3-32B. This suggests that distillation on a limited dataset may be insufficient to endow models with the robust, generalizable skills required for advanced visual-code synthesis. This result is in line with recent findings in knowledge distillation research, where it has been shown that dynamically adjusting the distillation data to focus on areas of large performance gaps between teacher and student models is more effective than using a static dataset (Liu et al., 2024b).

**Opportunities remain in challenging scenarios.** All models record their lowest scores on the Intensive Interactive cases within the static–dynamic classification tasks. They also perform worst on complex, project-level visualization scenarios—such as "Management System" cases—highlighting clear avenues for future improvement in these demanding settings.

**Generalist skills outperform specialist expertise.** Perhaps our most striking finding is that instruction-tuned generalist models outperform domain-specific counterparts. Specifically, Qwen-2.5-Instruct surpasses both the code-focused Qwen-2.5-coder and the vision-specialized Qwen2.5-VL-72B. This compellingly illustrates that producing high-quality visual artifacts is not a simple sum of isolated code generation and visual understanding abilities. Rather, it demands a higher-order synthesis of capabilities—including robust reasoning, nuanced instruction following, and an implicit sense of design aesthetics. These are precisely the holistic, meta-level skills that top-tier generalist models acquire through vast and diverse training, and which benchmarks like **Artifacts-Bench** are uniquely poised to evaluate.

**Visual and Code-Based Analysis** We categorize the 10 checklist items into two groups based on their evaluation dependencies: 5 vision-oriented checklists and 5 code-oriented checklists. As shown in Figure 4, The results demonstrate a positive correlation between vision and code scores, suggesting that model improvements tend to be comprehensive rather than isolated to specific capabilities. Focusing solely on visual scores or interactive experience may overlook critical strengths or weaknesses in the underlying code generation. Conversely, exclusive attention to code quality risks missing important visual aspects, leading to incomplete assessments. Furthermore, as model capabilities advance, they increasingly prioritize the visual presentation of generated code, thereby enhancing real-world usability.

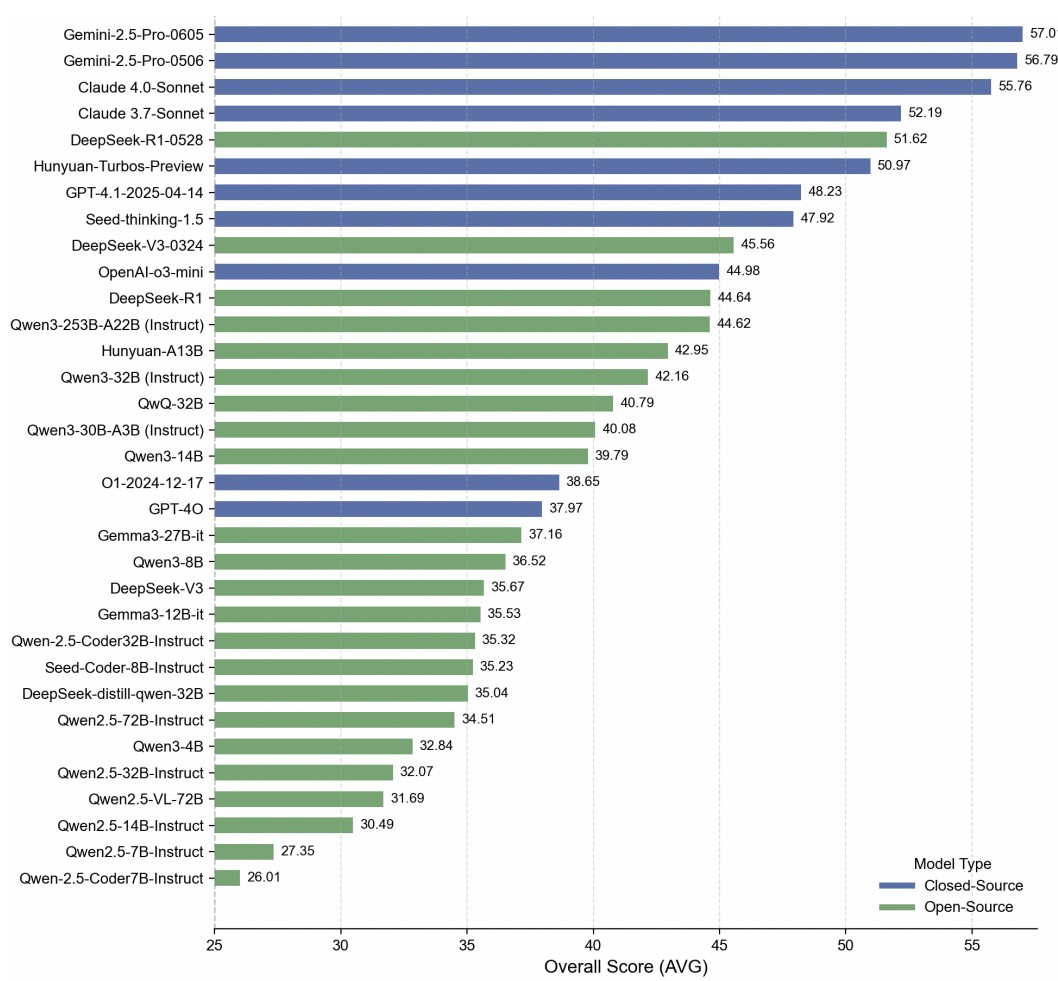

Figure 9: An overview of the competitive landscape on ArtifactsBench, scored by the Gemini-2.5-Pro-0506 referee. This chart summarizes the overall scores (AVG) of leading models, highlighting the current state-of-the-art and the performance distribution across different model families.

**Difficulty Analysis**    We split the benchmark into three tiers of increasing difficulty. As illustrated in Figure 5, even the best-performing models struggle to surpass 50 points on the most challenging subset, indicating that our benchmark remains far from saturation. Moreover, the models' relative rankings remain consistent across all tiers, and each tier continues to offer strong discriminative power—demonstrating that our benchmark reliably differentiates model capabilities at every level of difficulty.

## A.5 VALIDATION WITH HUMAN EXPERTS

To validate the fidelity of our automated MLLM-based evaluation, we conduct a rigorous human evaluation study. We randomly selected 280 queries, along with the corresponding data from 6 models, and have them independently scored by multiple engineers with extensive front-end development experience. The process follows a **double-blind protocol**: annotators remain unaware of the MLLM's scores, and the samples appear in randomized order to mitigate bias. The final human ground-truth score represents the median of the individual annotators' ratings.

To quantify the agreement between our automated referee and human experts, we compute the pairwise consistency rate, denoted as **Pair ACC**. For a given query with $m$ model responses, we can form $\frac{m(m-1)}{2}$ unique pairs of responses. We then count the number of pairs for which the MLLM referee and the human judges agree on the rank ordering (i.e., which response is better). The con-

Table 6: Checklist phrasing robustness for representative models on ArtifactsBench. "Original" uses the default checklist; "Paraphrased" uses semantically equivalent paraphrases; "Order+Template" additionally shuffles item order and switches to an alternative rubric template. Scores are global averages. Variations are minor and do not change the induced ranking or partial-order constraints.

| Model | Original | Paraphrased | Order+Template |
|---|---|---|---|
| Claude Sonnet 4 (20250514) | 57.28 | 57.51 | 57.43 |
| DeepSeek-V3-0324 | 43.50 | 43.58 | 43.54 |
| DeepSeek-R1 | 41.41 | 41.33 | 41.51 |
| Qwen3-235B-A22B | 41.09 | 41.26 | 41.22 |
| Claude 3.5 Sonnet (20241022) | 39.85 | 40.15 | 39.76 |
| GPT-4o | 33.54 | 33.71 | 33.55 |

sistency rate is the ratio of these concordant pairs to the total number of pairs. This metric allows us to select the most reliable MLLM referee and robustly demonstrates the strong correlation between our automated evaluation and expert human judgment.

Beyond referee–human agreement, we also assess inter-annotator reliability and the structure of disagreements. In total, 13 annotators with front-end engineering experience participate: 12 provide blind scores and 1 performs spot checks and, when necessary, requests re-annotation. For each of the 6 models on 280 queries, we aggregate multiple generations into a single candidate per (model, query) pair (via majority voting over sampled outputs), yielding 1,680 instances to rate. Annotators use a 6-level scale (0–5, where 0 denotes severely unacceptable and 5 denotes fully satisfactory with possible extra credit) and see all 6 model outputs for a given query side-by-side without model identities; ties are further broken by pairwise ranking among equal-scored outputs. To estimate label variance, each (model, query) instance is independently scored by three annotators, and we compute standard multi-rater agreement metrics (e.g., Fleiss' $\kappa$, Krippendorff's $\alpha$) as well as summary statistics indicating substantial internal consistency.

We additionally compare the empirical score distributions of human judges and the automated referee (after normalizing its scores to the same 0–5 scale). Human scores are distributed as: 0 (2.75%), 1 (18.37%), 2 (37.40%), 3 (29.79%), 4 (10.76%), 5 (0.918%); the normalized automatic scores follow: 0 (0.036%), 1 (29.00%), 2 (46.33%), 3 (17.86%), 4 (3.03%), 5 (0.206%). Two patterns emerge: (i) humans are stricter on clearly bad cases (assigning more 0s), while the automatic referee is more conservative at the lowest bin, and (ii) discrepancies concentrate on highly interactive, complex tasks, whereas mid- and low-performing regions show closely matched distributions. Together with the high Pair ACC, these statistics suggest that our MLLM-as-Judge closely tracks expert preferences overall while exposing systematic, interpretable residual differences.

## A.6 ROBUSTNESS TO CHECKLIST PHRASING AND TEMPLATE VARIANTS

Because our fine-grained checklists are LLM-drafted and then human-refined, a natural concern is that the referee might overfit to specific wording or template structure. To stress-test robustness, we perform two perturbation experiments on a subset of strong and mid-range models (Claude Sonnet 4 (20250514), DeepSeek-V3-0324, DeepSeek-R1, Qwen3-235B-A22B, Claude 3.5 Sonnet (20241022), GPT-4o). For each query, we (i) generate multiple paraphrases of the original checklist items using a high-capacity LLM (GPT-OSS-120B), preserving semantics while varying phrasing; and (ii) randomly shuffle item order and switch from a "question + scoring rule" template to an alternative "expected behavior + penalty conditions" template. The perturbed checklists are manually spot-checked to ensure semantic equivalence before being used to re-score all 1,825 queries.

Table 6 reports the global average scores under the original checklist, paraphrased checklist, and the order/template-perturbed checklist. We observe that absolute score changes are very small (typically < 0.3 points on the 0–100 scale), and, more importantly, the induced rankings and partial-order constraints over these models remain unchanged; only a few tie-breaks among very close systems flip under perturbation. Given the size of ArtifactsBench and the density of per-query scoring, these results suggest that our MLLM-as-Judge is reasonably robust to superficial checklist phrasing and template variations, and that the benchmark is not overly sensitive to any particular wording choice.

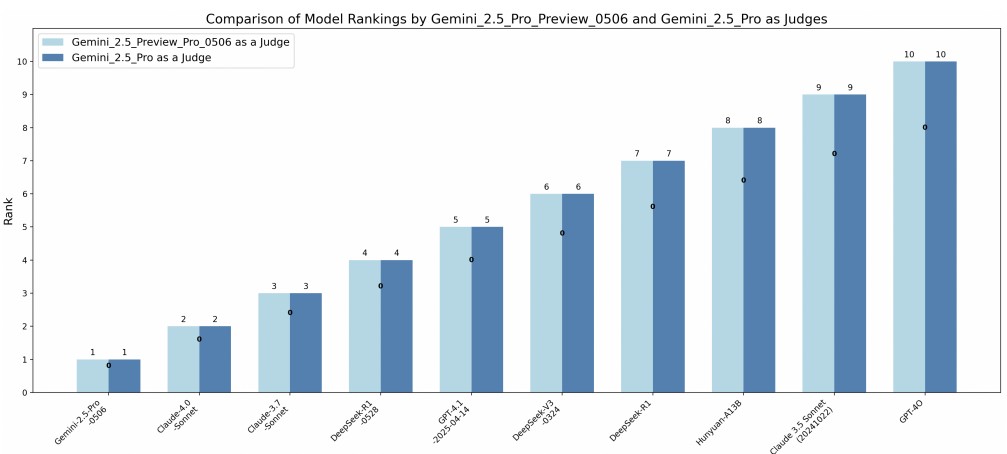

Figure 10: Version stability: Gemini-2.5-Pro-Preview-0506 vs. stable Gemini-2.5-Pro. The induced partial-order constraints are identical, preserving leaderboard conclusions.

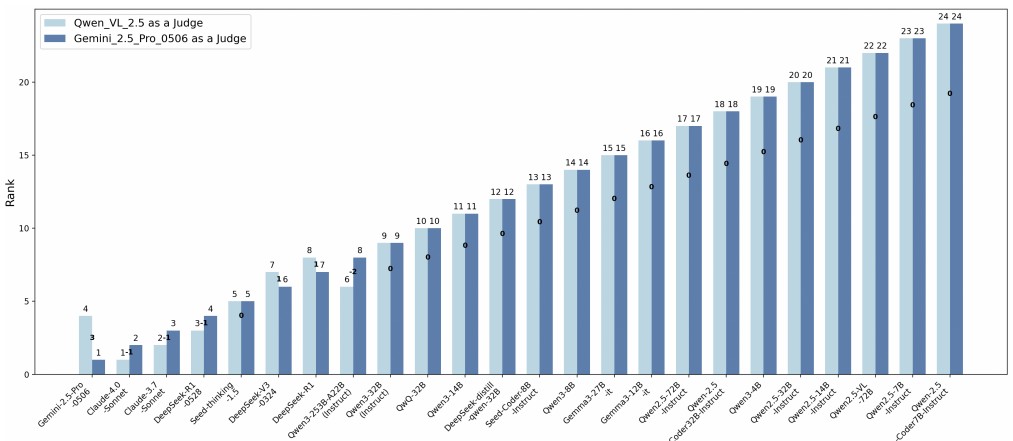

Figure 11: Cross-judge comparison (Gemini-2.5-Pro-Preview-0506 vs. Qwen2.5-VL-72B): highly consistent partial orders; minor top-tier differences.

**Cross-judge ranking consistency**  We further compare the partial-order ranking induced by two state-of-the-art referee models: the closed-source Gemini-2.5-Pro-Preview-0506 and the open-source Qwen2.5-VL-72B. As illustrated in Figure 11, the two referees yield largely consistent order constraints over model performance. Residual differences are concentrated among top-tier systems, where the stronger referee exhibits finer discriminative power. This observation supports two conclusions: (1) our findings are robust to the choice of MLLM referee, and (2) stronger referees provide higher resolution at the head of the leaderboard without altering the overall competitive landscape.

**Version stability of the Gemini referee**  Given that Gemini-2.5-Pro-Preview-0506 has been deprecated, we additionally compare it with the stable Gemini-2.5-Pro release. As shown in Appendix Figure 10, the two versions induce *identical partial-order constraints* over model rankings on **ArtifactsBench**. This means every pairwise ordering decision is the same, and replacing the preview with the stable version does not change any leaderboard conclusions.

**Ablation studies.**  We present ablation studies in Tables 4 and 5 to examine the alignment between different referee models and human judgments. The experimental configurations include: (1) "w/o img" - inputting only the query and answer to the LLM without images; (2) "w/ img" - leveraging the model's visual capability by providing the query, answer, and one execution screenshot; (3) "w/ imgs" - extending the "w/ img" configuration with multiple screenshots to capture dynamic visual effects; (4) "w/ caption" - replacing images with MLLM-generated descriptions; and (5)

Table 7: Average token usage per query on ArtifactsBench for representative models. **Question**, **Checklist**, and **Code** denote text tokens for the task description, checklist, and model answer, respectively (computed with the Qwen2.5 tokenizer). **Text-Total** is the sum of these three components. **Visual** counts image tokens from three execution screenshots (259 tokens per image for Gemini-2.5-Pro, totaling 777 tokens). **MLLM_Judge_Tokens** is the combined token budget seen by the referee.

| Model | Question | Checklist | Code | Text-Total | Visual | MLLM_Judge_Tokens |
|---|---|---|---|---|---|---|
| GPT-4o | 100.77 | 917.15 | 853.80 | 1871.72 | 777 | 2648.72 |
| GPT-OSS-120B | 100.77 | 917.15 | 2643.10 | 3661.02 | 777 | 4438.02 |
| DeepSeek-V3-0324 | 100.77 | 917.15 | 2290.37 | 3308.29 | 777 | 4085.29 |
| Claude 3.5 Sonnet | 100.77 | 917.15 | 1145.37 | 2163.29 | 777 | 2940.29 |
| Claude 3.7 Sonnet | 100.77 | 917.15 | 2621.14 | 3639.06 | 777 | 4416.06 |
| Claude 4 Sonnet | 100.77 | 917.15 | 4568.65 | 5903.48 | 777 | 6680.48 |
| GPT-4.1 | 100.77 | 917.15 | 1673.25 | 2691.68 | 777 | 3468.68 |

"only w/ imgs"/"only w/ caption" - removing the answer from the "w/ imgs" and "w/ caption" input configurations respectively.

Table 4 reveals two key findings: First, the significant improvement in pair accuracy when including execution screenshots demonstrates that multimodal LLMs effectively utilize visual information for more reasonable evaluation. Second, comparing columns 2 and 3 shows that multiple screenshots help models capture dynamic effects, further enhancing prediction accuracy. However, since referee models can extract additional strengths and weaknesses from the code-level perspective, their evaluation criteria may diverge from human judgments. This explains why even the strongest referee model, Gemini, shows some discrepancy in scoring consistency with human assessments, which simultaneously demonstrates ArtifactsBench's advantage of comprehensively evaluating answers from multiple dimensions.

The first three rows of Table 5 reveal that describing execution screenshots as text inputs can improve the evaluation accuracy of large models. Nevertheless, when the model itself has strong visual analysis capabilities, directly inputting screenshots yields better performance. The last two rows of Table 5 indicate that evaluation effectiveness decreases when only providing the query and execution screenshots, confirming the necessity of including the answer in the input. In addition, we provide a detailed analysis of the consistency analysis between ArtifactsBench and WebDev Arena in the appendix.

### A.7 COMPUTATIONAL COST OF VISUAL EVIDENCE

To make the trade-off between visual information and computational budget explicit, we quantify the token usage induced by our screenshots. Text tokens for questions, checklists, and model answers are counted with the `Qwen2.5` tokenizer. For images, we follow the accounting of the Gemini-2.5-Pro referee, where each screenshot of our standardized resolution corresponds to **259** image tokens; with three screenshots per query, this totals **777** visual tokens per evaluation. All statistics are averaged over the 1,825 queries in ArtifactsBench.

Table 7 reports the average per-query token budget for a set of representative models when three screenshots are provided to the referee. For GPT-4o, GPT-4.1, Claude and other strong baselines, visual tokens account for a substantial fraction of the Multimodal-Judge input. Using GPT-4o as a concrete reference, adding one screenshot to the text-only configuration increases the token budget by **13.88%**, while providing three screenshots increases it by **41.51%**. Together with the ablation results in Table 4, which show that Pair ACC rises from **79.06%** (no images) to **87.10%** (one image) and **90.95%** (three images), these numbers motivate our choice of three staged screenshots as a near-saturated operating point. A further increase to five screenshots improves agreement only slightly to **91.2%**, while raising the token cost by roughly **20%** relative to the three-screenshot setting, indicating diminishing returns.

### A.8 PROGRAMMATIC HTML/DOM QUALITY CHECKS

In response to requests for non-LLM, fully programmatic checks (e.g., DOM assertions), we conducted a preliminary study using a lightweight static HTML analyzer, which assigns each model

Table 8: Programmatic HTML quality scores (DOM_SCORE) vs. ArtifactsBench scores for 30+ LLMs. DOM_SCORE is computed from static HTML along five dimensions (Structure, Accessibility, Semantics, SEO, Best Practices; weights 0.25/0.30/0.20/0.15/0.10). Values are tightly clustered (mostly > 95), indicating that basic HTML quality is nearly saturated and offers little discriminative power compared to ArtifactsBench.

| MODEL | DOM_SCORE | | | | | | ArtifactsBench_SCORE |
|---|---|---|---|---|---|---|---|
| | Structure | Accessibility | Semantics | SEO | Best Practices | AVG | |
| *Open-Source Large Language Models* | | | | | | | |
| Qwen2.5-7B-Instruct | 98.7 | 98.5 | 99.7 | 88.5 | 99.8 | 97.4 | 27.35 |
| Qwen2.5-14B-Instruct | 98.6 | 98.3 | 99.8 | 87.1 | 99.8 | 97.1 | 30.49 |
| Qwen2.5-32B-Instruct | 98.8 | 98.4 | 99.8 | 88.2 | 99.8 | 97.4 | 32.07 |
| Qwen2.5-72B-Instruct | 98.7 | 98.4 | 99.7 | 89.1 | 99.8 | 97.5 | 34.51 |
| Qwen2.5-VL-72B | 98.8 | 98.2 | 99.9 | 85.0 | 99.8 | 96.9 | 31.69 |
| Qwen-2.5-Coder7B-Instruct | 98.7 | 98.2 | 99.8 | 82.3 | 99.8 | 96.4 | 26.01 |
| Qwen-2.5-Coder32B-Instruct | 98.4 | 98.2 | 99.7 | 89.2 | 99.8 | 97.4 | 35.32 |
| QwQ-32B | 98.8 | 97.3 | 99.7 | 77.9 | 99.7 | 95.5 | 40.79 |
| Qwen3-4B | 99.2 | 97.7 | 99.9 | 82.5 | 99.8 | 96.4 | 32.84 |
| Qwen3-8B | 99.2 | 97.3 | 99.8 | 80.5 | 99.8 | 96.0 | 36.52 |
| Qwen3-14B | 99.1 | 97.6 | 99.8 | 83.1 | 99.8 | 96.4 | 39.79 |
| Qwen3-32B (Instruct) | 99.2 | 97.5 | 99.8 | 82.7 | 99.8 | 96.4 | 42.16 |
| Qwen3-30B-A3B (Instruct) | 99.3 | 97.5 | 99.8 | 82.4 | 99.8 | 96.3 | 40.08 |
| Hunyuan-A13B | 97.8 | 97.7 | 99.4 | 86.4 | 99.7 | 96.6 | 42.95 |
| Qwen3-253B-A22B (Instruct) | 98.9 | 97.5 | 99.7 | 82.7 | 99.8 | 96.3 | 44.62 |
| DeepSeek-distill-qwen-32B | 98.8 | 97.4 | 99.7 | 84.4 | 99.8 | 96.5 | 35.04 |
| Seed-Coder-8B-Instruct | 98.8 | 97.4 | 99.7 | 84.4 | 99.8 | 96.5 | 35.23 |
| Gemma3-12B-it | 98.8 | 98.6 | 99.8 | 81.0 | 99.7 | 96.3 | 35.53 |
| Gemma3-27B-it | 98.9 | 98.3 | 99.7 | 78.8 | 99.7 | 95.9 | 37.16 |
| DeepSeek-V3 | 98.6 | 98.6 | 99.8 | 88.3 | 99.8 | 97.4 | 35.67 |
| DeepSeek-R1 | 98.8 | 97.0 | 99.7 | 80.5 | 99.8 | 95.8 | 44.64 |
| DeepSeek-V3-0324 | 97.7 | 97.5 | 99.4 | 88.1 | 99.7 | 96.7 | 45.56 |
| DeepSeek-R1-0528 | 97.0 | 96.5 | 98.6 | 90.0 | 99.7 | 96.4 | **51.62** |
| *Closed-Source Large Language Models* | | | | | | | |
| Seed-thinking-1.5 | 98.8 | 97.3 | 99.7 | 84.4 | 99.8 | 96.5 | 47.92 |
| GPT-4o | 98.9 | 98.4 | 99.9 | 88.4 | 99.8 | 97.5 | 37.97 |
| GPT-4.1-2025-04-14 | 98.7 | 98.5 | 99.7 | 82.4 | 99.8 | 96.5 | 48.23 |
| O1-2024-12-17 | 100.0 | 100.0 | 100.0 | 80.4 | 100.0 | 97.0 | 38.65 |
| OpenAI-o3-mini | 100.0 | 96.0 | 100.0 | 89.5 | 100.0 | 97.2 | 44.98 |
| Hunyuan-Turbos-Preview | 97.4 | 96.2 | 97.4 | 91.5 | 99.8 | 96.4 | 50.97 |
| Claude 3.7-Sonnet | 97.5 | 97.1 | 99.5 | 88.3 | 99.6 | 96.6 | 52.19 |
| Claude 4.0-Sonnet | 97.0 | 96.3 | 99.8 | 90.3 | 99.8 | 96.4 | 55.76 |
| Gemini-2.5-Pro-Preview-0506 | 97.9 | 98.1 | 99.6 | 90.2 | 99.7 | 97.3 | 56.79 |
| Gemini-2.5-Pro-Preview-0605 | 98.0 | 98.2 | 99.6 | 90.3 | 99.7 | 97.4 | **57.01** |

output a DOM_SCORE in $[0, 100]$. The tool parses the rendered HTML and evaluates five dimensions: (i) *Structure* (duplicate IDs, excessive empty tags, overuse of inline styles); (ii) *Accessibility* (alt text for images, label–form association, descriptive button/link text); (iii) *Semantics* (use of semantic tags such as header/nav/main/..., avoidance of div/span overuse, basic table structure); (iv) *SEO* (presence and quality of title, meta description, lang, viewport, and heading structure); and (v) *Best Practices* (number and placement of external scripts, deprecated tags, missing image dimensions). Each dimension is scored and linearly combined with fixed weights (accessibility 30%, structure 25%, semantics 20%, SEO 15%, best practices 10%) to obtain the final DOM_SCORE.

Applied to all 30+ models in Table 3, we find that DOM_SCORE is universally high and tightly clustered: almost all systems score above 95 out of 100 with very small variance, and the induced ranking has negligible discriminative power compared to our main ArtifactsBench scores. This suggests that modern LLMs rarely fail at basic HTML syntax, structural correctness, or coarse accessibility/SEO hygiene, and that static DOM quality is no longer the primary bottleneck. In contrast, our checklist-guided, MLLM-as-Judge protocol—which jointly inspects code behavior and multi-step visual evidence—exposes large performance gaps on whether the generated artifacts actually satisfy the specified visual layout and interaction logic. In practice, we therefore use DOM_SCORE as a sanity check for grossly malformed outputs, while relying on ArtifactsBench scores for meaningful model comparison.

Table 9: Reproducibility of referee scores on ArtifactsBench. Each model is scored five times by Gemini-2.5-Pro with different random seeds (fixed model outputs and rendering). "avg" denotes the mean score and "var" the empirical variance.

| Model | infer1 | infer2 | infer3 | infer4 | infer5 | avg | var |
|---|---|---|---|---|---|---|---|
| Claude Sonnet 4 (20250514) | 57.28 | 57.13 | 57.09 | 56.88 | 57.09 | 57.09 | 0.0210 |
| DeepSeek-V3-0324 | 43.50 | 43.23 | 43.51 | 43.36 | 43.69 | 43.46 | 0.0304 |
| DeepSeek-R1 | 41.41 | 41.21 | 41.60 | 41.39 | 41.54 | 41.43 | 0.0228 |
| Qwen3-235B-A22B | 41.09 | 40.75 | 40.98 | 40.85 | 41.01 | 40.94 | 0.0180 |
| Claude 3.5 Sonnet (20241022) | 39.85 | 40.19 | 40.04 | 39.74 | 39.86 | 39.94 | 0.0319 |
| GPT-4o | 33.54 | 33.31 | 33.64 | 33.75 | 33.71 | 33.59 | 0.0306 |

## A.9 SCORE REPRODUCIBILITY STUDY

To empirically quantify score variation under repeated evaluation, we conduct a small reproducibility study on six representative models (Claude Sonnet 4 (20250514), DeepSeek-V3-0324, DeepSeek-R1, Qwen3-235B-A22B, Claude 3.5 Sonnet (20241022), GPT-4o). For each model, we fix the model outputs and rendering stack, and re-run the Gemini-2.5-Pro referee five times with different random seeds. Table 9 reports the per-run scores, their average, and the empirical variance. The observed variances are extremely small (on the order of $10^{-2}$ on a 0–100 scale), indicating that ArtifactsBench scores are highly stable across independent referee runs for strong-performing systems.

## A.10 REPRODUCIBILITY AND FUTURE DIRECTIONS

We release dataset specs, checklist templates, evaluation scripts, and referee settings; future work targets richer dynamics beyond discrete screenshots, agentic multi-turn self-debugging, and broader artifact domains.

## A.11 EVALUATION RESULTS FROM ADDITIONAL SCORING REFEREES

Table 10 and 11 present the detailed evaluation results from two distinct MLLM-as-a-Judge referees, Gemini-2.5-Pro and Qwen2.5-VL-72B, respectively, assessing multiple mainstream large language models on the ArtifactsBench benchmark. Gemini-2.5-Pro, as a leading closed-source multimodal model, provides a high-accuracy benchmarking standard, while the open-source Qwen2.5-VL-72B offers a more cost-effective alternative.

The results demonstrate a high degree of consistency in the model rankings computed by these two referee models, despite differences in their architectures and capabilities. This consistency not only validates the stability of the ArtifactsBench evaluation metrics but also indicates a convergent consensus among multimodal experts in judging code generation quality. Nevertheless, minor discrepancies are observed in the fine-grained scoring of certain complex interactive tasks, reflecting potential cognitive biases in how different MLLMs interpret long-horizon interaction logic and dynamic visual feedback.

Table 10: Detailed evaluation results on ArtifactsBench, scored by the Gemini-2.5-Pro referee. Performance is detailed across interactivity levels (**SV**: Static Visual, **MMD**: Mild-to-Moderate Dynamics, **HD**: High Dynamics, **II**: Intensive Interactive) and task categories (**GAME**, **SVG**, **WEB**, **SI**: Simulation, **MS**: Management System). **AVG** is the global average. **IFLEN** represents the average response length. Top proprietary multimodal models lead overall, and performance scales with model capacity.

| MODEL | IFLEN | SCORE | | | | | | | | | |
|---|---|---|---|---|---|---|---|---|---|---|---|
| | | SV | MMD | HD | II | GAME | SVG | WEB | SI | MS | AVG |
| *Closed-Source Large Language Models* | | | | | | | | | | | |
| GPT-5 | – | 72.24 | 79.82 | 75.17 | 69.81 | 77.89 | 73.40 | 71.31 | 79.41 | 64.95 | 72.55 |
| Claude-opus-4-1 | – | 58.07 | 60.47 | 61.35 | 59.42 | 61.63 | 57.03 | 60.11 | 58.87 | 57.43 | 59.76 |
| Gemini-2.5-Pro | – | 60.14 | 59.18 | 58.71 | 55.62 | 58.38 | 65.33 | 58.12 | 55.54 | 53.18 | 57.74 |
| Claude Sonnet 4 (20250514) | – | 56.82 | 60.06 | 60.08 | 55.16 | 57.98 | 53.85 | 58.36 | 58.35 | 55.38 | 57.28 |
| o3-2025-04-16 | – | 54.90 | 56.88 | 55.92 | 51.85 | 54.33 | 56.37 | 52.95 | 55.75 | 50.21 | 54.04 |
| Claude 3.7 Sonnet (20250219) | 15480.06 | 49.76 | 51.64 | 53.17 | 50.79 | 51.11 | 45.37 | 53.52 | 50.81 | 51.74 | 51.32 |
| GPT-4.1-2025-04-14 | 7290.94 | 43.81 | 47.35 | 47.28 | 45.92 | 49.30 | 42.47 | 46.11 | 45.39 | 41.05 | 45.95 |
| Claude 3.5 Sonnet (20241022) | 6391.96 | 41.44 | 42.08 | 42.40 | 36.95 | 38.46 | 41.94 | 41.26 | 39.43 | 38.17 | 39.85 |
| GPT-4o | 4882.36 | 34.91 | 33.59 | 35.74 | 31.30 | 33.04 | 33.75 | 34.22 | 31.44 | 32.10 | 33.54 |
| *Open-Source Large Language Models* | | | | | | | | | | | |
| GPT-OSS-120B | 16018.79 | 58.11 | 56.78 | 58.90 | 54.93 | 53.88 | 54.19 | 58.77 | 57.69 | 56.97 | 56.91 |
| Qwen3-235B-A22B-Thinking-2507 | 34357.84 | 53.63 | 55.66 | 56.90 | 54.32 | 56.35 | 44.80 | 55.90 | 57.35 | 54.09 | 55.01 |
| GLM-4.5 | 21854.10 | 51.07 | 54.94 | 53.10 | 49.68 | 54.79 | 51.79 | 51.66 | 52.06 | 47.30 | 51.33 |
| Qwen3-235B-A22B-Instruct-2507 | 20765.47 | 48.35 | 50.37 | 53.03 | 50.16 | 50.67 | 40.41 | 52.19 | 50.24 | 50.83 | 50.62 |
| GLM-4.5_Air | 20925.02 | 48.26 | 52.53 | 51.70 | 46.44 | 52.79 | 48.41 | 49.70 | 55.60 | 44.40 | 48.90 |
| DeepSeek-R1-0528 | 19215.71 | 48.11 | 53.32 | 49.54 | 45.45 | 50.46 | 45.06 | 47.86 | 54.08 | 42.69 | 47.73 |
| Kimi K2 Instruct | 7116.99 | 50.11 | 51.28 | 49.88 | 44.31 | 47.08 | 50.61 | 46.81 | 48.88 | 46.15 | 47.65 |
| Qwen3-Coder-480B-A35B-Instruct | 17581.53 | 46.32 | 50.77 | 48.68 | 45.99 | 49.27 | 40.18 | 48.11 | 49.66 | 46.06 | 47.15 |
| DeepSeek-V3-0324 | 11443.06 | 44.04 | 43.47 | 46.82 | 40.95 | 45.29 | 40.20 | 45.56 | 37.22 | 42.17 | 43.50 |
| DeepSeek-R1 | 10751.63 | 42.99 | 43.75 | 43.69 | 38.68 | 40.89 | 42.43 | 41.91 | 40.80 | 39.82 | 41.41 |
| Qwen3-235B-A22B | 19314.92 | 42.75 | 42.03 | 43.01 | 38.76 | 40.68 | 40.15 | 42.62 | 38.92 | 39.39 | 41.09 |
| hunyuan-A13B | 17924.89 | 41.09 | 41.73 | 42.14 | 39.94 | 40.84 | 39.87 | 42.34 | 37.35 | 40.27 | 40.95 |
| KAT-V1-40B | 23262.57 | 34.34 | 37.67 | 38.01 | 33.32 | 33.42 | 28.20 | 35.84 | 37.17 | 35.78 | 35.21 |

Table 11: Evaluation results on ArtifactsBench using Qwen2.5-VL-72B as the MLLM referee. The table presents model rankings based on comprehensive assessment across various dimensions. **RANK** indicates the position in evaluation order, **AVG RESPONSE LENGTH** represents the average length of model responses, and **QWEN2.5-VL-72B SCORE** shows the evaluation score when using Qwen2.5-VL-72B as the multimodal referee.

| RANK | MODEL | INSTITUTION | AVG RESPONSE LENGTH | QWEN2.5-VL-72B SCORE |
|---|---|---|---|---|
| *Open-Source Large Language Models* | | | | |
| 1 | Qwen2.5 7B-Instruct | Qwen | 7905.21 | 42.72 |
| 2 | Qwen2.5 14B-Instruct | Qwen | 6334.34 | 44.76 |
| 3 | Qwen2.5 32B-Instruct | Qwen | 5115.49 | 46.09 |
| 4 | Qwen2.5 72B-Instruct | Qwen | 6029.47 | 51.30 |
| 5 | Qwen2.5-VL-72B | Qwen | 3539.15 | 44.45 |
| 6 | QwQ-32B | Qwen | 20232.53 | 60.41 |
| 7 | Qwen3-4B | Qwen | 35479.79 | 48.11 |
| 8 | Qwen3-8B | Qwen | 22319.97 | 56.29 |
| 9 | Qwen3-14B | Qwen | 15118.26 | 59.97 |
| 10 | Qwen3-32B (Instruct) | Qwen | 17394.15 | 63.14 |
| 12 | Qwen3-30B-A3B (Base) | Qwen | 35679.24 | 23.43 |
| 13 | Qwen3-253B-A22B (Instruct) | Qwen | 19400.61 | 66.35 |
| 14 | Qwen-2.5-Coder7B-Instruct | Qwen | 5800.23 | 34.57 |
| 15 | Qwen-2.5-Coder32B-Instruct | Qwen | 6318.59 | 49.72 |
| 16 | DeepSeek-R1 | DeepSeek | 10754.69 | 66.22 |
| 17 | DeepSeek-R1-0528 | DeepSeek | 20780.42 | 73.78 |
| 18 | DeepSeek-V3-0324 | DeepSeek | 11455.42 | 66.27 |
| 19 | DeepSeek-distill-qwen-32B | DeepSeek | 9249.36 | 57.14 |
| 20 | Gemma3-12B-it | Google | 7955.42 | 52.49 |
| 21 | Gemma3-27B-it | Google | 7912.14 | 52.99 |
| 22 | Seed-Coder-8B-Instruct | ByteDance | 8934.07 | 56.73 |
| *Closed-Source Large Language Models* | | | | |
| 23 | Seed-thinking-1.5 | ByteDance | 14823.72 | 68.74 |
| 24 | Claude 3.7 | Anthropic | 15470.66 | 73.80 |
| 25 | Claude 4.0-Sonnet | Anthropic | 20633.88 | 78.86 |
| 26 | Gemini-2.5-Pro-0506 | Google | – | 71.01 |

## A.12 QUALITY FILTERING OF QUERIES

We use the following prompt to filter the quality of queries, with the aim of selecting high-quality, practical, complete, and privacy-free queries.

---

You are a professional Query Evaluation Expert with advanced analytical reasoning abilities.
Your task is to conduct a comprehensive, step-by-step evaluation of a given question using a detailed Chain of Thought (CoT) approach.

Evaluation Framework:

Stage 1: Comprehensive Problem Understanding
  - Carefully analyze the original problem statement
  - Identify explicit and implicit requirements
  - Assess technical complexity and contextual constraints

Stage 2: Please rate the given question based on the following five dimensions and provide the rating result:
  - Quality: Does the question have a clear logical structure, is it expressed accurately, and does it avoid ambiguity?
  - Creativity: Does the question offer novelty, providing new perspectives or problem-solving opportunities compared to existing ones?
  - Relevance: Does the question have practical value, such as applicability to specific use cases or its ability to assess valuable knowledge points or skills?
  - Completeness: Is the question description clear and comprehensive, with no critical information missing? Are the given conditions sufficient and reasonable to support the derivation of a correct answer?
  - Privacy: Does the question avoid requesting or involving any sensitive personal information, such as phone numbers, addresses, or other identifiable details?

Output Specifications:
  - Rating Range:
      - Each evaluation dimension and the overall score range from 1 to 10, with 1 being the worst and 10 being the best.
  - Reasoning: All scores must have clear, specific reasoning to support them.
  - Structure: The final output must be clear, concise, and professional.
  - Objectivity: The rating must be neutral and fair.
  - The final output should be a JSON object with the five scores and an overall score (average of the five dimensions), as shown below:

```json
{
   "Quality": "8",
   "Creativity": "7",
   "Relevance": "9",
   "Completeness": "9",
   "Privacy": "7",
   "Total Score": "8.0"
}
```

Please rate the following question according to the above standards:

———question-start———
Question
———question-end———

---

Figure 12: The prompt for query quality filtering.

Table 12: ROUGE-L based contamination analysis following the prefix-completion protocol of ConceptMath (Wu et al., 2024b). "Fully contaminated" corresponds to using the ground-truth suffix as prediction given the prefix (upper bound 1.0). All tested models obtain much lower scores, suggesting limited verbatim contamination.

| Model | ROUGE-L |
|---|---|
| Fully contaminated upper bound | 1.00000 |
| Qwen2.5-7B-Instruct | 0.07679 |
| Qwen2.5-14B-Instruct | 0.08350 |
| Qwen2.5-32B-Instruct | 0.08703 |
| Qwen3-8B | 0.03595 |
| Qwen3-32B | 0.02964 |
| Seed-Coder-8B-Instruct | 0.08523 |
| GPT-OSS-120B | 0.05884 |

## A.13 CONTAMINATION DETECTION PROTOCOL

To further audit potential data contamination, we adopt a model-based recall analysis inspired by the protocol of ConceptMath (Wu et al., 2024b). Concretely, for each benchmark query we feed the *first half* of the question text to several representative LLMs (Qwen2.5/3, Seed-Coder-8B, GPT-OSS-120B) and ask them to complete the *second half*. We then compute the ROUGE-L similarity between the generated suffix and the ground-truth suffix, and average over all queries. As an upper bound, we also compute a *fully contaminated* scenario in which the ground-truth suffix is used directly as the "prediction," yielding ROUGE-L = 1.0 by construction. Lower scores therefore indicate weaker lexical overlap and less evidence of exact memorization.

Table 12 reports the resulting ROUGE-L scores. All evaluated models lie in the range 0.03–0.09, far below the fully contaminated upper bound of 1.0, and different models exhibit heterogeneous and relatively low recall rates. Combined with our multi-stage deduplication and source curation pipeline described in Sec. 2, these results suggest that large-scale, verbatim contamination of ArtifactsBench in current LLM training corpora is unlikely, although we do not claim the absence of any partial or semantic overlap.

## A.14 CLASSIFICATION OF QUERIES

We use Gemini-2.5-Pro to classify queries, and the specific prompt is shown in Figures 13 and 14. They only require a query as input. For interaction-level stratification, we first ask a high-capacity LLM to assign each query a discrete interaction score using the prompt in Figure 14, then bucket scores into four levels (Static Visual, Mild-to-Moderate Dynamics, High Dynamics, Intensive Interactive) and finally conduct human review to correct residual errors, resulting in an estimated $\sim 96\%$ agreement between automatic labels and annotators. The resulting distribution over the 1,825 tasks is 396/117/536/776 for Static Visual, Mild-to-Moderate Dynamics, High Dynamics, and Intensive Interactive, respectively.

## A.15 THE PROMPT FOR MODEL SCORING

We present the prompt for generating the checklist in Figures 15 and 16, which takes a query as input. After manual review, it will be put into use. The final scoring prompt, shown in Figure 17, requires the query, answer, and checklist as input.

## A.16 API ENDPOINTS FOR PROPRIETARY MODELS

For reproducibility, we list here the public API endpoints used to access proprietary models in our experiments:

You are a master at categorizing queries into specific classes. You will receive a series of queries, and your task is to accurately assign them to predefined major and minor categories based on their content. The output format for each query should be "MajorCategory-MinorCategory".

Evaluation Framework:
Phase 1: Comprehensive Understanding of the Problem
   - Carefully analyze the original problem statement
   - Identify explicit and implicit requirements
   - Assess technical complexity and contextual constraints

Phase 2: Based on the query content, refer to the following classification structure for judgment. If the result falls under "Other," specify the minor category and output it in the required format:

1. **Game Development**:
   - Puzzle | Sports | Shooting | Casual | Strategy
   - Simulation/Management | Role-Playing | Adventure | Action/Rhythm

2. **Web Applications**:
   - Communication | Online Shopping | Education/Learning | Blogs/Forums | Web Visuals

3. **Management Systems**:
   - Frontend/Backend Platforms | File Management | Hardware Management

4. **Multimedia Editing**:
   - Image Editing | Audio Editing | Video Production

5. **Data Science**:
   - Data Visualization Dashboards | Statistical Analysis | Predictive Modeling | Machine Learning

6. **Simulation & Modeling**:
   - Physics Simulation | Mathematical Abstraction | 3D Simulation

7. **SVG Generation**:
   - SVG Icons/Logos | SVG Images | SVG Posters

8. **Mermaid Flowcharts**:
   - Code Flowcharts | Logic Flowcharts | Mind Maps

9. **Other**

The final output should be a JSON object containing the category, as shown below:
```json
{
    "Class": "Game Development-Strategy"
}
```

Please categorize the following question based on the above criteria:

——question-start——
Question
——question-end——

Figure 13: The prompt for category classification.

You are a master proficient in assigning queries to specific categories. You will receive a series of queries, and based on their content, you must accurately classify them into the predefined categories below and output the results.

Evaluation Framework:
Phase 1: Comprehensive Understanding of the Problem
    • Carefully analyze the original problem statement
    • Identify explicit and implicit requirements
    • Assess technical complexity and contextual constraints

Phase 2: Based on the query content, refer to the following classification structure for judgment and output the results in the specified format:
1. Static Visual Class (Class: 0 points):
    • Definition: Visualization is required, but only a single screenshot is needed for static judgment, without the need for dynamic effects or manual interaction.
    • Typical features: Front-end code must be written for visualization, including verbs like "display, show, illustrate" but without dynamic modifiers.

2. Dynamic Visual Class (Class: 1-10 points):
    • Definition: Visualization requires changes across time and space dimensions.
    • Grading criteria:
        • Mildly Dynamic (1-3 points): Basic animation effects
            Examples: "changes over time," "simple transitions"
        • Moderately Interactive (4-6 points): Parameter adjustments
            Examples: "adjust X-axis range," "click a button," "drag a slider"
        • Highly Interactive (7-10 points): Multimodal real-time operations
            Examples: "rotate a 3D model," "input an account," "upload a file"

The final output should be a JSON object containing the category score, as shown below:
```json
{
"Class": "3"
}
```
Please classify the following question based on the above criteria:

——question-start——
Question
——question-end——

Figure 14: The prompt for visual classification.

You are a senior and meticulous code review expert, proficient in multiple programming languages, front - end technologies, and interaction design. Your task is to generate a check - list for the received [Query]. The responses to the [Query] mainly include source code (in multiple programming languages), algorithm implementation, data structure design, system architecture diagrams, front - end visualization code (such as HTML/SVG/JavaScript), descriptions of interaction logic, and related technical explanations, with a primary focus on front - end visualization. Please use your code knowledge and aesthetic experience to modify the following check - list, and the full score should be 100 points.

Role Positioning
• Responsibility: Like an authoritative technical review committee member in the industry, you must be objective, comprehensive, and unbiased.
• Attitude: Meticulous, professional, and uncompromising, good at identifying various details and potential risks.
• Others: Possess high aesthetic talent, with excellent aesthetics and high requirements for user experience.

Example:
Query:
You are a code expert. Please use your professional knowledge to generate accurate and professional responses. Note to ensure that the generated code can be executed and displayed as much as possible. Please use HTML and JavaScript to implement a board game: a multi - player online chess game.
Task: Design a multi - player online chess game system that allows players to play against each other over the network and save the game progress.
Hint: You can use server - side synchronization to manage the game state and design a reconnection mechanism.
Checklist:
1. Is the chess game combat system fully implemented?
• Review whether the code accurately implements the chessboard coordinate system through HTML/JavaScript, and whether it includes collision detection for piece movement and validation of legal moves (including special rules such as castling/en passant). Score 0 if the core interaction logic is not implemented, 5 if only basic movement is implemented, and 10 if all international chess rules are fully included.
2. Is the player online combat function implemented?
• Check whether the WebSocket implementation includes a heartbeat mechanism, a packet verification sequence, and automatic degradation on disconnection (transfer to local temporary storage). Two - way state verification between the front - end and back - end is required. Deduct 5 points if the retransmission mechanism is missing, and 3 points if network latency compensation is not handled. The full score is 10 points.
3. Is the server - side synchronization mechanism designed and a reconnection function provided?
• Evaluate whether the server synchronization strategy uses differential incremental synchronization instead of full - scale updates, and whether an operation prediction mechanism is adopted. Two - way verification of client prediction and server correction is required. Deduct 5 points if the state drift exceeds 200ms. Check whether a disconnection reconnection mechanism is designed to ensure that players can resume the game after being disconnected. The full score is 10 points.
4. Is the complete game lifecycle management constructed?
• Check whether the code includes complete game lifecycle management, including state management such as game pause/resume, multi - game history backtracking, and spectator mode. Deduct 5 points if game serialization storage is not implemented, and 3 points if the crash recovery mechanism is missing. Give 10 points if fully implemented.

Figure 15: The first part of the prompt for visual classification

5. Is the code robust?

• Evaluate whether the code can handle common abnormal situations (such as out - of - bounds input, network interruption, user operation errors, etc.) and provide friendly error prompts or recovery mechanisms. Code with strong robustness should be able to effectively handle these edge cases, giving 10 points. If the robustness is average, give 5 points, and if no exceptions are handled, give 0 points.

6. Are there any innovative features that are eye - catching?

• Check whether the code includes surprise features that enhance the experience (e.g., 1. Real - time AI move scoring 2. Exporting game recordings with commentary 3. Interactive bullet screens for friends watching). Add 3 points for each practical innovative feature implemented (maximum 10 points).

7. Are there any redundant features?

• Strictly check three types of redundancy: 1. Redundant implementation of similar functions (e.g., multiple undo logics coexisting) 2. Function modules unrelated to chess (e.g., a built - in music player) 3. Fancy effects that affect performance (e.g., particle explosion animations). Deduct 3 points for each redundancy found, and directly deduct 10 points if the core functions are interfered with by redundant code.

8. Does the code have engineering quality?

• Review modular design (such as separating game logic/view/network layers), unit test coverage, and build process automation. Deduct 5 points if global state pollution is found or design patterns are not used; deduct 5 points if the code duplication rate is too high (over 30%); deduct 5 points if the build process is not automated. The full score is 10 points.

9. Does the interface vision meet professional design standards?

• Evaluate whether the overall design follows modern design principles: 1) Harmonious color matching (no more than 3 primary colors) 2) Proper layout spacing (element spacing follows the 8px multiple principle) 3) Professional font system (body font size $\geq$ 14px, line height over 1.5 times). Deduct 3 points for each crowded visual element, 5 points for a glaring color combination, and 5 points for chaotic text - image layout. The full score is 10 points.

10. Is the dynamic interaction smooth and seamless?

• Judge whether the dynamic effects conform to human perception characteristics: 1) Click feedback delay $\leqslant$ 100ms 2) Transition animation duration controlled between 300 - 500ms 3) Clear visual focus guidance. Deduct 5 points for each operation without feedback, 3 points for visual after - images during fast sliding, and 5 points for hard - to - find key function buttons. The full score is 10 points.

• I hope you can modify items 1 - 4 according to the [Query] I give you. The other items can be fine - tuned but try to be consistent with the example I provided. Note that each item should be judged in combination with screenshots as much as possible. There must be 10 items. Ensure that the detection difficulty of the check - list is high and the requirements are relatively strict. The final output should be a complete check - list wrapped in a JSON block, without including any other content. Refer to the following example:

```json
{
    "checklist": ¡specific checklist¿
}
```

Please generate the checklist for the following query according to the above standards:

———Query starts———
Query
———Query ends———

Figure 16: The second part of the prompt for visual classification.

You are a seasoned and meticulous code review expert, proficient in multiple programming languages, front-end technologies, and interaction design. Your task is to conduct an in-depth analysis and scoring of the received [question] and [answer]. The [answer] may include source code (in various programming languages), algorithm implementations, data structure designs, system architecture diagrams, front-end visualization code (such as HTML/SVG/CSS/JavaScript), interaction logic descriptions, and related technical explanations. Please leverage your coding expertise and aesthetic experience to thoroughly examine the [answer] content from the following dimensions and provide scores along with detailed review comments. You should be very strict and cautious when giving full marks for each dimension.

Role Definition

Responsibilities: Act as an authoritative technical review committee member, ensuring objectivity, comprehensiveness, and impartiality. Attitude: Rigorous, professional, and unsparing, adept at identifying details and potential risks.
Additional Traits: Possess exceptional aesthetic talent, with high standards for visual appeal and user experience.
I have only extracted the last segment of HTML or SVG code from the provided answer for visualization. The content is adaptively scrolled to capture the entire page.

**Scoring Criteria:**

$Checklist

- The final output should be a JSON object containing the dimensions above, following this example:
```json
{
    "Overall Score": "35"
}
```
Reason:...

Please score the following question according to the standards above:

——Problem starts——
$Question
——Problem ends——

——Answer starts——
$Answer
——Answer ends——

Figure 17: The final scoring prompt.

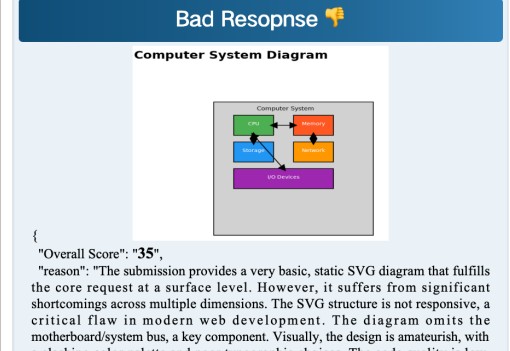

Figure 18: Visualization Results of SVG Poster Category Test Cases.

- **Hunyuan-TurboS-Preview**: https://cloud.tencent.com/document/product/1729/105701#4.-.E7.A4.BA.E4.BE.8B
- **Seed-thinking-1.5**: https://www.volcengine.com/product/doubao
- **GPT series**: https://openai.com/api/
- **Claude series**: https://www.claude.com/platform/api
- **Gemini series**: https://ai.google.dev/gemini-api/docs/models

## B  VISUALIZATION RESULTS

We present a portion of the evaluation dataset from ArtifactsBench, along with some reasoning processes, scores, prompts, and screenshots (if three are identical, only one is shown), as illustrated in Figures 18–24.

**● ● ● Prompt**

>_ You are a code expert. Please use your professional knowledge to generate accurate and professional responses. Make sure the code you generate is executable whenever possible. The classic brick-breaking game code can be run in a browser.

**Good Resopnse 👍**

Classic Brick Breaker | Classic Brick Breaker | Classic Brick Breaker

```
{
  "Overall Score": "48",
  "reason": "The provided code is a solid, functional, and well-commented 'first-pass' implementation of a Brick Breaker game. It successfully sets up the core gameplay loop, including paddle control, ball movement, and basic collision detection. The visual presentation is clean and aesthetically pleasing with a consistent retro-neon theme.\\n\\nHowever, it scores low in several key areas that define a complete and polished game experience.\\n- **Gameplay Depth:** It completely lacks difficulty progression, multiple levels, power-ups, or special brick types. The game is a single, static experience.\\n- **Physics and Collision:** The physics model is overly simplistic. The ball's bounce angle off the paddle is not variable, and more importantly, the brick collision logic is flawed, only allowing for vertical bounces regardless of where the ball hits the brick. This leads to unrealistic gameplay.\\n- **Code Quality:** From an engineering standpoint, the reliance on numerous global variables for game state management is a significant weakness that hinders scalability and maintainability.\\n- **User Experience:** While visually clean, the complete absence of audio feedback detracts from the immersive experience. The game is also not robust against browser resizing.\\n\\nIn summary, it's an excellent demonstration of how to use HTML Canvas for a simple game but stops short of being a fully-featured or robust game application. It serves as a great starting point but requires significant additions to be considered a complete product."
}
```

**Bad Resopnse 👎**

Brick Breaking Game | Brick Breaking Game | Brick Breaking Game

```
{"Overall Score": 29,
  "Review": "The provided code successfully implements a rudimentary version of a brick-breaking game. It's a classic example often used in introductory programming tutorials. While it demonstrates the basic concepts—rendering on a canvas, handling user input, and a simple game loop—it lacks the refinement, robustness, and depth expected of a complete game. Its primary value is educational, showcasing how to get started, but it is not a polished product. The physics are simplistic and buggy, the code architecture is non-existent, and the user experience is jarring. It serves as a functional foundation but would require a complete rewrite to reach a professional standard."
}
```

Figure 19: Visualization Results of Game Development Category Test Cases.

**● ● ● Prompt**

>_ You are a code expert. Please use your professional knowledge to generate accurate and professional responses. Make sure the code you generate is executable for demonstration purposes. Write a piece of frontend code that simulates the motion of the sun as a celestial body.

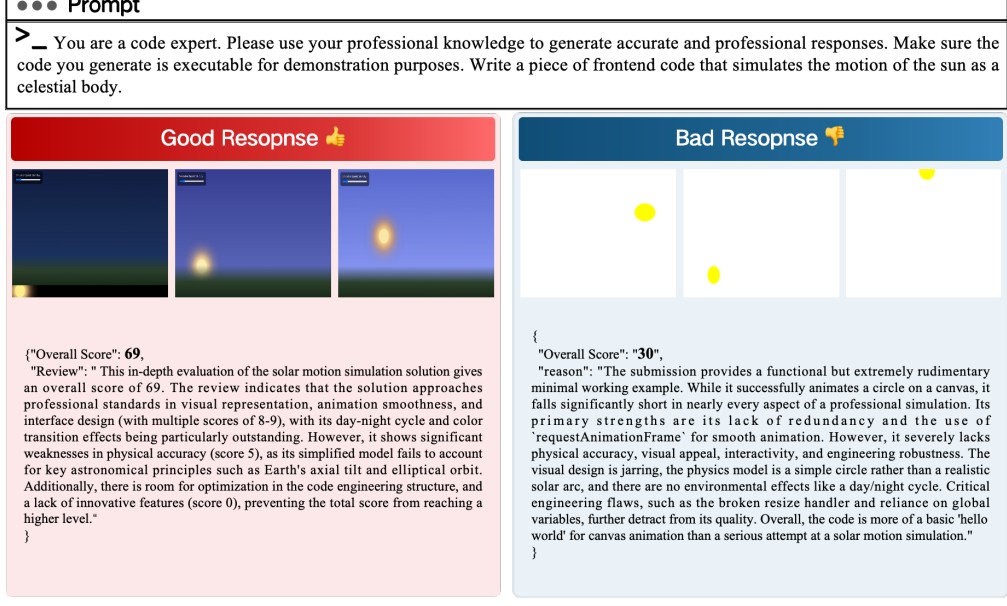

**Good Resopnse 👍**

```
{"Overall Score": 69,
  "Review": " This in-depth evaluation of the solar motion simulation solution gives an overall score of 69. The review indicates that the solution approaches professional standards in visual representation, animation smoothness, and interface design (with multiple scores of 8-9), with its day-night cycle and color transition effects being particularly outstanding. However, it shows significant weaknesses in physical accuracy (score 5), as its simplified model fails to account for key astronomical principles such as Earth's axial tilt and elliptical orbit. Additionally, there is room for optimization in the code engineering structure, and a lack of innovative features (score 0), preventing the total score from reaching a higher level."
}
```

**Bad Resopnse 👎**

```
{
  "Overall Score": "30",
  "reason": "The submission provides a functional but extremely rudimentary minimal working example. While it successfully animates a circle on a canvas, it falls significantly short in nearly every aspect of a professional simulation. Its primary strengths are its lack of redundancy and the use of `requestAnimationFrame` for smooth animation. However, it severely lacks physical accuracy, visual appeal, interactivity, and engineering robustness. The visual design is jarring, the physics model is a simple circle rather than a realistic solar arc, and there are no environmental effects like a day/night cycle. Critical engineering flaws, such as the broken resize handler and reliance on global variables, further detract from its quality. Overall, the code is more of a basic 'hello world' for canvas animation than a serious attempt at a solar motion simulation."
}
```

Figure 20: Visualization Results of Network Application Category Test Cases.

● ● ● **Prompt**

>_ You are a code expert. Please use your professional knowledge to generate accurate and professional responses. Make sure the generated code is executable for display. Please help me create HTML code for the Google Chrome dinosaur game.

| Good Resopnse 👍 | Bad Resopnse 👎 |
|---|---|

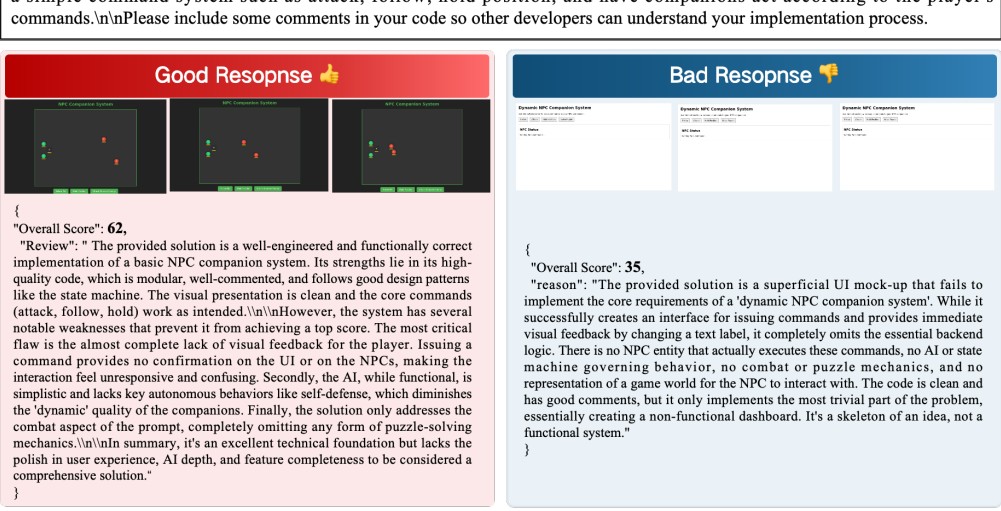

{
"Overall Score": **30**,
 "Review": " An in-depth analysis of the provided HTML code for a Dinosaur Game reveals a rudimentary implementation that, while functional at a surface level, falls significantly short of the \"professional\" and \"complete\" standard it claims. The code successfully establishes a basic interactive loop but omits numerous core features and engineering best practices essential for a polished and robust game."
}

{
"Overall Score": **20**,
 "reason": "An in-depth analysis of the provided code for a Google Chrome dinosaur game reveals a rudimentary implementation that captures the most basic concept but falls short on nearly every technical and design metric expected of a professional-quality product. The code serves as a functional-but-flawed starting point, demonstrating fundamental logic but ignoring crucial aspects of game development such as performance, robustness, and user experience."
}

Figure 21: Visualization Results of Game Development Category Test Cases.

● ● ● **Prompt**

>_ You are a code expert. Please use your professional knowledge to generate accurate, expert responses. Make sure the code you generate is executable.\nPlease use HTML and JavaScript to implement a dynamic NPC companion system.\n\n**Task**: Implement a dynamic NPC companion system where players can command teammates in combat or puzzle-solving.\n\n**Hint**: You can set up a simple command system such as attack, follow, hold position, and have companions act according to the player's commands.\n\nPlease include some comments in your code so other developers can understand your implementation process.

| Good Resopnse 👍 | Bad Resopnse 👎 |
|---|---|

{
"Overall Score": **62**,
 "Review": " The provided solution is a well-engineered and functionally correct implementation of a basic NPC companion system. Its strengths lie in its high-quality code, which is modular, well-commented, and follows good design patterns like the state machine. The visual presentation is clean and the core commands (attack, follow, hold) work as intended.\\n\\nHowever, the system has several notable weaknesses that prevent it from achieving a top score. The most critical flaw is the almost complete lack of visual feedback for the player. Issuing a command provides no confirmation on the UI or on the NPCs, making the interaction feel unresponsive and confusing. Secondly, the AI, while functional, is simplistic and lacks key autonomous behaviors like self-defense, which diminishes the 'dynamic' quality of the companions. Finally, the solution only addresses the combat aspect of the prompt, completely omitting any form of puzzle-solving mechanics.\\n\\nIn summary, it's an excellent technical foundation but lacks the polish in user experience, AI depth, and feature completeness to be considered a comprehensive solution."
}

{
"Overall Score": **35**,
 "reason": "The provided solution is a superficial UI mock-up that fails to implement the core requirements of a 'dynamic NPC companion system'. While it successfully creates an interface for issuing commands and provides immediate visual feedback by changing a text label, it completely omits the essential backend logic. There is no NPC entity that actually executes these commands, no AI or state machine governing behavior, no combat or puzzle mechanics, and no representation of a game world for the NPC to interact with. The code is clean and has good comments, but it only implements the most trivial part of the problem, essentially creating a non-functional dashboard. It's a skeleton of an idea, not a functional system."
}

Figure 22: Visualization Results of Game Development Category Test Cases.

● ● ● **Prompt**

> _ You are a code expert. Please use your professional knowledge to generate accurate and professional responses. Make sure the generated code is executable whenever possible. Please help me create a small program that can analyze the asymptotic time complexity of input C language program fragments (note that these are incomplete program fragments) and output the results in Big O notation. Please generate this as an HTML program.

**Good Resopnse** 👍

{
"Overall Score": **45,**
  "Review": " The submission presents a visually polished and user-friendly web application with a clean interface and smooth interactions. The developer clearly understands front-end design principles and has done an excellent job in creating an aesthetically pleasing and intuitive UI. The accompanying text, which explains the methodology and candidly states the tool's limitations, is also a mark of professionalism.\n\nHowever, the core functionality—the actual complexity analysis—is critically flawed. The analysis algorithm fails to distinguish between sequential and nested loops, a fundamental concept in complexity analysis, leading to incorrect results for all but the simplest, single-loop or perfectly-nested-loop code fragments. While the heuristic, regex-based approach is acceptable for a \"small program,\" the implementation of the analysis logic itself is incorrect. In essence, the project is a well-designed facade for a non-functional analysis engine."
}

**Bad Resopnse** 👎

{
"Overall Score": **34,**
  "reason": "An in-depth analysis of the provided solution reveals significant shortcomings in its core functionality, robustness, and design. The approach of using regular expressions for syntactic analysis is fundamentally flawed and brittle, leading to incorrect assessments for any code that deviates even slightly from a few hardcoded patterns. While the submission is a self-contained, executable HTML file, it fails to deliver a reliable or useful tool for its stated purpose."
}

Figure 23: Visualization Results of Data Science Category Test Cases.

● ● ● **Prompt**

> _ You are a code expert. Please use your professional knowledge to generate accurate and professional responses. Make sure the code you generate is executable for demonstration. Create a web module that includes factory area dropdown buttons, department filter keys, cost code filter keys, date range filter keys, and the ability to import security personnel, security unit prices, annual/monthly/daily security expenses, and annual/monthly/daily security budget.

**Good Resopnse** 👍

{
"Overall Score": **52,**
  "Review": " Based on the in-depth analysis, the solution scores 52/100. It successfully implements the core filtering features (department, cost code, date range) and daily expense calculations with smooth interaction. However, it fails critically on data import functionality, has a poorly rendered and unprofessional interface, uses a non-dynamic dropdown and a single-file code structure with weak error handling, and lacks required monthly/annual calculations and any innovative elements."
}

**Bad Resopnse** 👎

{
"Overall Score": **41,**
  "reason": " The interaction design has significant flaws. Firstly, there are no transition animations; when filters are applied, the data table content changes abruptly, which is not a smooth experience (-3 points). More critically, the data import process uses a blocking `alert()` for feedback, which is a dated and disruptive UI pattern. This is a poor choice for user interaction and feedback (-5 points). The combination of abrupt updates and disruptive alerts makes the interaction feel unpolished."
}

Figure 24: Visualization Results of Management System Category Test Cases. Although the answer on the left scored lower in aesthetics, it ultimately received a higher overall score because it is more functional than the answer on the right.