# OpenReview forum: "ArtifactsBench: Bridging the Visual-Interactive Gap in LLM Code Generation Evaluation"
_ICLR.cc/2026/Conference — Submitted to ICLR 2026_

### Official Review · Reviewer_e67X · 2025-10-26

**Soundness:** 2
**Presentation:** 2
**Contribution:** 3
**Rating:** 6
**Confidence:** 4

**Summary:**

The paper proposes a benchmark to evaluate LLMs' visual artifact generation capabilities, using multimodal LLMs as judges and task-specific checklists. The benchmark covers both static and dynamic tasks, along with an evaluation framework to judge both code and visual components. Separate studies are conducted to assess the alignment between the proposed evaluation and human experts.

**Strengths:**

- The paper is well-structured, and the motivation is sound.
- The benchmark covers a wide range of tasks.
- The data creation and evaluation pipeline has sufficient human grounding, which improves (e.g., task filtering and checklist generation) and makes the benchmark robust (e.g., alignment study of evaluation scores on 280 instances with human experts).
- A large number of open-source and closed-source LLMs are evaluated across different model scales and capabilities to provide a holistic overview of current models on artifact generation tasks.

**Weaknesses:**

In general, the work lacks the following:
- crucial details about the dataset creation pipeline, which either creates confusion or gives an incomplete picture
- insights or takeaways on how to improve the current LLMs for these tasks (e.g., the paper only provides which tasks are challenging, but through the checklist, the discussion could be more detailed)

These weaknesses are supported by the questions below. I will be happy to increase the scores if my concerns are adequately addressed.

**Questions:**

- The paper talks about reproducible scores - is it implicitly assumed because the MLLM-judge has to score using the checklist? It would be better to conduct a small study on the variation in scores, at least for the responses of strong-performing LLMs.
- Can you describe, out of 1825 tasks, how many are static and how many are dynamic?
- Although a high-level difference between some existing related benchmarks is provided (e.g., Table 2), individual differences are not clear:
  - How is CHA measured in Table 2 for all benchmarks (in the paper, only ArtifactsBench and WebBench CHA are studied)
  - Why is the CHA assigned only "Mid" and "High" scores?
  - What are the benefits of ArtifactsBench over FullFront (the rows are almost identical in the Table)?
  - Why is DOM alignment (can you also briefly describe this evaluation?) not sufficient for evaluation, e.g., WebBench?
  - Is the output from the tasks in these previous benchmarks identical to the outputs expected from your tasks?
- Why are three screenshots enough (in the text, it is described as before, during, and after, but why is just one screenshot for "during" enough)? Does it mean that each dynamic task implicitly assumes a single interaction/transition? If so, why can't simpler methods be designed for evaluating such dynamic tasks?
- Is Hunyuan-TurboS open-sourced or accessible through API?
- Within the dual-referee setup, the open-source and closed-source MLLM evaluations are not combined to give a final score. Is there any interaction between the two referees, or are the two referees used independently to provide aligned scores? If so, then the dual-referee setup is somewhat misleading.
- Could Gemini-2.5-Pro being used as a referee lead to the responses from the same model family being scored higher?
- **Formatting**: The captions of tables are inconsistently placed.
- Is there an intuition about how alarming a drop-off of 1 or 2 points is, i.e., between achieving 65 and 64, how bad (or on what aspects) is the second model worse than the first model?
- Is inclusion of visual and code-oriented scores beneficial? Figure 4 demonstrates a strong positive correlation between the two scores; it would have been interesting to see the need for both scores when one score is high, the other decreases, and highlights the deficiencies of the model; however, that not being the general case casts doubt on the checklist preparation and evaluation metric. In more detail, having only one of the two scores (e.g., code-oriented scores) would be enough, while a significant motivation of the benchmark and evaluation was to evaluate the visual component (e.g., visual-oriented scores).
- Are there overlaps between the tasks of ArtifactsBench and other benchmarks?
- Questions on benchmark creation pipeline:
  - What was the human effort required during the task creation (e.g., in hours for contamination control, prompt rewriting, difficulty calibration, and checklist refinement)?
  - In what aspect would the pipeline differ for different topics, e.g., for data science and multimedia editing?
  - I wanted to take a look at some sample examples of tasks, but was unable to download or view the dataset (waited for 5 minutes) at https://anonymous.4open.science/r/ArtifactsBench-F7F9/dataset/artifacts_bench.json (is this also happening for other reviewers?). Can you maybe provide 1 or 2 examples per topic in the appendix?
  - The input to the benchmark creation pipeline is not described, i.e., what does "candidates from expert showcases" mean? Can you provide examples?
  - How do you extract "prompts, checklists, and normalized DOM/CSS/JS" from the above sources?
  - Can you provide examples of what underspecification means within "ambiguity repair"?

---

> ### Author Response · Authors · 2025-11-19
>
> Thank you very much for your careful reading and for the many constructive questions. We appreciate the opportunity to clarify the data pipeline, reproducibility, and insights, and have significantly expanded the relevant parts of the manuscript in response (all new text is highlighted in red).
>
> - **A_for_Q1 (Q1/W1: Reproducible scores and variance study):**
>
>   You are right that simply stating “reproducible scores” is not sufficient without quantifying variance. In the revised manuscript we add a dedicated appendix subsection titled **“Score Reproducibility Study”**, where we evaluate score variation under repeated referee runs.
>
>   Concretely, we fix the model outputs and rendering stack, and re-run the Gemini-2.5-Pro referee five times with different random seeds for six representative models (Claude Sonnet 4, DeepSeek-V3-0324, DeepSeek-R1, Qwen3-235B-A22B, Claude 3.5 Sonnet, GPT-4o). The corresponding table reports per-run scores, their average, and empirical variance. Variances are extremely small (on the order of \(10^{-2}\) on a 0–100 scale), indicating that ArtifactsBench scores are highly stable across independent referee runs for strong-performing systems. This supports the claim that the reported leaderboards are both discriminative and reproducible.
>
> - **A_for_Q2 (Q2/W1: Static vs dynamic composition):**
>
>   We have made the static vs dynamic composition explicit in the main dataset section and the appendix subsection “Classification of Queries” (both with red text). Along the interaction axis, we stratify the 1,825 tasks into four classes: Static Visual, Mild-to-Moderate Dynamics, High Dynamics, and Intensive Interactive. The resulting distribution is 396 / 117 / 536 / 776 tasks, respectively.
>
>   If we coarsen this into “static vs dynamic,” then 396 tasks are purely static visual (single-screenshot judgment is sufficient), and the remaining 1,429 tasks involve non-trivial dynamics or interaction. We also describe the prompts and criteria used to assign these interaction levels in the appendix, where we show the classification prompts and report an estimated ∼96% agreement between automatic labels and annotators.
>
> - **A_for_Q3.1 (Q3.1: Definition of CHA in the comparison table):**
>
>   We clarify in the revised manuscript that CHA (human consistency) in the comparison table is an ordinal, qualitative indicator, not a new numerical metric we re-compute for all benchmarks. For ArtifactsBench and WebBench, we provide explicit quantitative evidence (Pair ACC vs human experts and rank consistency vs WebDev Arena); for the other benchmarks we rely on their published human studies or the nature of their supervision.
>
>   Specifically, we mark CHA as High when the benchmark is either directly defined by human judgments (e.g., WebDev Arena) or anchored in carefully curated, human-validated labels (e.g., HumanEval and SWE-Bench tests). We mark CHA as Mid when prior work reports reasonable but not near-perfect alignment with human preferences, or when evaluation is primarily proxy-based (e.g., DOM/pixel similarity) with limited evidence of large-scale human validation (WebBench, WebGen-Bench, WebChoreArena, FullFront). We have added this clarification around the comparison table in the “ArtifactsBench: A Benchmark for Visual Code Generation” section.
>
> - **A_for_Q3.2 (Q3.2: Why only “Mid” and “High” for CHA):**
>
>   In our taxonomy, the CHA column is intentionally coarse and ordinal (Low/Mid/High). In practice, none of the listed benchmarks is “anti-correlated” with human preferences to the extent that we would label it Low; instead, they range from moderately aligned (Mid) to strongly aligned (High). For example, our own analysis shows that WebBench’s model rankings achieve 69.4% normalized Footrule consistency with WebDev Arena, which is clearly better than random but significantly below ArtifactsBench’s 94.4%. We now explicitly explain this interpretation in the text accompanying the comparison table.

---

> > ### Author Response · Authors · 2025-11-19
> >
> > - **A_for_Q3.3 (Q3.3: Benefits over FullFront):**
> >
> >   FullFront is an important step toward evaluating web comprehension and generation, but it differs from ArtifactsBench in three key ways, which we now emphasize in the “Related Work” section. First, FullFront is largely model-synthesized and focuses on QA-style evaluation of the front-end development process, with only a small subset (≈50) of relatively simple text-to-code cases, mostly static HTML; ArtifactsBench, by contrast, centers on end-to-end artifact generation with 1,825 executable tasks across nine domains, including games, simulations, data dashboards, and management systems.
> >
> >   Second, our evaluation protocol is explicitly interaction-aware and visual: we execute code in a sandbox, capture staged screenshots, and use a checklist-guided MLLM-as-Judge to score visual fidelity, dynamics, interaction, and code quality. Third, we quantitatively demonstrate stronger alignment with human experts and WebDev Arena (in the “Experiments” section and the appendix), offering fine-grained diagnostics over interaction levels and topic categories. These differences are summarized in the revised comparison table and discussed in the updated related work paragraph on FullFront.
> >
> > - **A_for_Q3.4 (Q3.4: Why DOM alignment is not sufficient):**
> >
> >   DOM alignment and related static programmatic checks are valuable but fundamentally limited for interactive visual artifacts. They primarily verify structural correctness and basic accessibility/SEO hygiene, and cannot capture layout quality, overlapping elements, broken visual flow, or whether the specified interaction trajectory (e.g., multi-step workflows, dynamic dashboards, game mechanics) is actually realized.
> >
> >   In the revised appendix subsection “Programmatic HTML/DOM Quality Checks” and the associated table, we show that a static DOM_SCORE (based on structure, accessibility, semantics, SEO, best practices) is almost saturated: most models score above 95/100 with very small variance, and the induced ranking has negligible discriminative power compared to ArtifactsBench scores. This helps explain why WebBench, which leans heavily on DOM alignment, achieves only 69.4% rank consistency with WebDev Arena, whereas ArtifactsBench reaches 94.4%. Our evaluation protocol is designed to complement DOM checks by directly assessing the rendered visual and interactive behavior.
> >
> > - **A_for_Q3.5 (Q3.5: Whether outputs are identical across benchmarks):**
> >
> >   The outputs are not identical. Prior benchmarks such as Design2Code and WebBench mostly expect static pages or DOM structures aligned to given designs or scripts. In contrast, ArtifactsBench expects runnable, interactive artifacts whose behavior matches open-ended natural-language instructions (and optional reference images), spanning games, simulations, dashboards, management systems, multimedia tools, and SVG graphics. Evaluation is based on actual execution in a sandbox and multimodal scoring of visual dynamics and code quality, rather than static reconstruction alone. We emphasize this distinction in the “ArtifactsBench: A Benchmark for Visual Code Generation” and “Related Work” sections.
> >
> > - **A_for_Q4 (Q4/W1: Why three screenshots are used and whether one “during” screenshot is enough):**
> >
> >   We do not assume that each dynamic task has only a single interaction; rather, we approximate the interaction trajectory with three staged screenshots (before/during/after scripted interaction) for scalability. In the revised manuscript we provide a detailed ablation in the “Evaluation Methodology” section and the appendix subsection **“Computational Cost of Visual Evidence”**, comparing 0, 1, 3, and 5 screenshots.
> >
> >   Using Gemini-2.5-Pro as referee, Pair ACC with human experts rises from 79.06% (no images) to 87.10% (one screenshot) and 90.95% (three screenshots); moving from three to five screenshots yields only a marginal gain (≈91.2%) at roughly 20% additional token cost over the three-screenshot setting. We therefore adopt three staged screenshots as a near-saturated operating point on the cost–accuracy curve. We explicitly acknowledge in the “Limitations and Future Work” subsection that very long-horizon workflows and subtle physics/UX timing remain challenging and outline richer scripted interactions, longer trajectories, and video-based or DOM-level analyses as future extensions.
> >
> > - **A_for_Q5 (Q5: Accessibility of Hunyuan-TurboS):**
> >
> >   Hunyuan-TurboS is not open-sourced, but it is accessible via a public cloud API. In our experiments we interact with it exclusively through its official API, without any private or internal interfaces. To improve reproducibility, the revised appendix includes a subsection titled **“API Endpoints for Proprietary Models”**, where we list the public endpoints for Hunyuan-TurboS and other proprietary models (GPT, Claude, Gemini, Seed-thinking). This allows others to reproduce our settings or run additional baselines on the same versions.

---

> > > ### Author Response · Authors · 2025-11-19
> > >
> > > - **A_for_Q6 (Q6: Dual-referee setup and independence of referees):**
> > >
> > >   We apologize for any ambiguity in the wording. In our dual-referee setup, the two referees (Gemini-2.5-Pro and Qwen2.5-VL-72B) are used independently; we do not combine their scores into a single aggregate metric, nor do they interact during scoring. Each referee evaluates all 1,825 tasks separately, producing its own leaderboard and score matrix (the main text reports results for Gemini-2.5-Pro, and the appendix provides full results and a separate leaderboard for Qwen2.5-VL-72B).
> > >
> > >   The purpose of the dual-referee design is to provide both (i) a strong, high-capacity proprietary referee for maximum accuracy and (ii) a fully reproducible open-source referee for the community. In the revised text and appendix (particularly in the cross-judge comparison figures) we explicitly analyze the partial-order constraints induced by both referees and show that they are highly consistent, with residual differences concentrated among top-tier models. We now clarify in the “Evaluation Methodology” and “Experiments” sections that “dual-referee” means “two parallel, transparent evaluation channels,” not “a fused score.”
> > >
> > > - **A_for_Q7 (Q7: Potential bias toward Gemini-family models):**
> > >
> > >   We considered the possibility that using Gemini-2.5-Pro as referee might systematically favor models from the same family. Empirically, we do not observe evidence of such bias. First, when we switch the referee from Gemini-2.5-Pro-Preview-0506 to the stable Gemini-2.5-Pro release, the induced partial-order constraints over model rankings remain identical (as shown in the “version stability” figure), indicating that Gemini version changes do not reorder the competitive landscape.
> > >
> > >   Second, when we use Qwen2.5-VL-72B as the referee instead (see the detailed Qwen2.5-VL-72B leaderboard and the cross-judge comparison figure in the appendix), the relative ranking of Gemini models versus other families remains consistent with the Gemini-based referee, rather than being substantially inflated only under the Gemini referee. Finally, our evaluation pipeline is checklist-driven and code-aware: each task has a task-specific 10-dimension checklist, and the referee receives detailed scoring criteria alongside the question, code, and screenshots, which constrains the evaluation and reduces the room for model-family-specific preference to dominate.
> > >
> > > - **A_for_Q8 (Q8: Inconsistent caption placement):**
> > >
> > >   We have globally reviewed all tables and figures and standardized caption placement in the revised manuscript: **table captions are placed above the tables and figure captions below the figures**. This change is mentioned in the ethics/fairness/licensing subsection and implemented consistently throughout the camera-ready draft.
> > >
> > > - **A_for_Q9 (Q9: Interpretation of 1–2 point differences):**
> > >
> > >   Our overall score is the sum of 10 checklist dimensions, each scored on a 0–10 scale, normalized to a 0–100 scale. A 1–2 point difference in the global score typically corresponds to either a clear improvement on one checklist dimension (e.g., from partially satisfying to fully satisfying a functional or visual requirement) or smaller, cumulative improvements across several dimensions. It is therefore noticeable but not dramatic—interpretable as “slightly better overall artifact quality.”
> > >
> > >   In the “Experiments” section and the appendix subsection **“Detailed Analysis of the Results”**, we illustrate how such differences manifest in practice. For mid-tier models with similar averages (e.g., several models in the mid-30s), differences tend to concentrate on static visual and simpler dynamic tasks; for top-tier models (e.g., in the mid-50s), the same 1–2 point differences are mainly driven by performance on High Dynamics and Intensive Interactive tasks. This suggests that as models improve, the same numeric gap increasingly reflects harder, more interactive scenarios.

---

> > > > ### Author Response · Authors · 2025-11-19
> > > >
> > > > - **A_for_Q10 (Q10/W2: Usefulness of vision vs code-oriented scores):**
> > > >
> > > >   You are correct that the figure relating vision- and code-oriented scores shows a strong positive correlation, which we see as evidence that capability improvements are holistic rather than siloed. However, we find that separating these two groups of dimensions is still useful for diagnosis. In the “Evaluation Methodology” section and the appendix subsection **“Visual and Code-Based Analysis”**, we discuss cases where one dimension is high and the other lags—e.g., visually appealing but brittle code (poor robustness, redundancy, poor engineering hygiene), or logically solid implementations with suboptimal layout, aesthetics, or feedback.
> > > >
> > > >   Moreover, our DOM_SCORE study in the appendix subsection “Programmatic HTML/DOM Quality Checks” shows that purely static HTML/DOM metrics are almost saturated and offer little discrimination, reinforcing the value of code-level checklist dimensions that go beyond syntax (robustness, scalability, modularity, redundancy). In practice, users can aggregate the 10 dimensions into a single score for ranking, but the separate vision/code splits help reveal why a model performs the way it does and guide targeted improvement (e.g., focusing on UX and visual design vs. robustness and engineering).
> > > >
> > > > - **A_for_Q11 (Q11/W1: Overlaps with other benchmarks):**
> > > >
> > > >   ArtifactsBench does not intentionally reuse tasks from existing benchmarks, and we apply multi-stage de-duplication and contamination control at the prompt, checklist, DOM/CSS/JS, and screenshot levels (as detailed in the dataset section and the appendix subsection **“Detailed Introduction to Data Collection and Cleaning”**). In addition, the contamination detection experiment described in the appendix subsection “Contamination Detection Protocol” (prefix-completion with ROUGE-L) suggests that large-scale verbatim overlap with current LLM training corpora is unlikely.
> > > >
> > > >   At the level of high-level categories (e.g., “web applications,” “SVG generation”), there is inevitably conceptual overlap with prior work, but the concrete tasks, prompts, and checklists in ArtifactsBench are newly constructed and curated. We explicitly avoid importing existing benchmark instances and aim instead to cover a broader, interaction-focused spectrum of real-world artifact types.
> > > >
> > > > - **A_for_Q12.1 (Q12.1/W1: Human effort in task creation):**
> > > >
> > > >   We now provide a more explicit account of human effort in the appendix subsections “Detailed Introduction to Data Collection and Cleaning” and “Validation with Human Experts” (red text). Roughly, prompt rewriting and quality checking involved about 2 people over 3 days; difficulty calibration combined multi-model performance with 2 people over 2 days of manual auditing and adjustment; checklist refinement required about 6 people over 3 days to curate and calibrate a 10% seed set. The 280-instance expert study involved 13 annotators over approximately 5 days. We believe these investments provide a reasonable balance between scalability and human grounding.
> > > >
> > > > - **A_for_Q12.2 (Q12.2/W1: Topic-specific differences in the pipeline):**
> > > >
> > > >   The overall pipeline stages are shared across topics, but certain steps are topic-aware. For example, prompt templates and quality filters differ between data-science dashboards and multimedia editing tasks; the checklists emphasize different dimensions (e.g., chart correctness and interactivity for data science, timeline/controls and playback behavior for multimedia). In the appendix, we include topic-specific prompt examples for classification and checklist generation, and our few-shot checklist seeds are stratified by topic to ensure that topic-specific expectations are reflected in the rubrics.
> > > >
> > > > - **A_for_Q12.3 (Q12.3/W1: Examples per topic and dataset access):**
> > > >
> > > >   In response to your request, we added a “Visualization Results” section in the appendix (red text), showing representative rendered examples from several topics, including SVG posters, game development, web applications, management systems, and data-science dashboards (Figures labelled “Visualization Results of ... Category Test Cases”). These examples are drawn directly from ArtifactsBench, not toy cases. We also checked and kept the anonymous repository online, and point readers to the dataset/artifacts_bench.json file and associated scripts there; we apologize if network latency prevented you from downloading it during review.

---

> > > > > ### Author Response · Authors · 2025-11-19
> > > > >
> > > > > - **A_for_Q12.4 (Q12.4/W1: Meaning of “candidates from expert showcases”):**
> > > > >
> > > > >   By “expert showcases” we refer to collections of front-end and visualization examples created by experienced practitioners, such as course project galleries, competition problems, curated blog demos, and high-quality CodePen/Observable snippets. In the revised appendix subsection **“Detailed Introduction to Data Collection and Cleaning”**, we give concrete examples of these sources and describe how we normalize them into text prompts and checklists (e.g., extracting the task description, interaction logic, and key visual requirements from an existing demo before rewriting it into an instruction suitable for LLMs).
> > > > >
> > > > > - **A_for_Q12.5 (Q12.5/W1: Extracting prompts, checklists, DOM/CSS/JS):**
> > > > >
> > > > >   For benchmark construction, we primarily need the textual prompt and the expected behavioral/visual constraints. From source artifacts, we first summarize the intended functionality and interaction in natural language (prompt) and then generate task-specific checklists using the checklist-generation pipeline described in the appendix (where we show prompt templates for checklist generation), followed by human refinement on a seed set. During evaluation, we extract code from model outputs using robust regular expressions (released scripts such as code_parser.py and extract_ans.py in the anonymous repository), and from the rendered page we obtain normalized DOM/CSS/JS for de-duplication and sanity checks as detailed in the dataset section and the appendix.
> > > > >
> > > > > - **A_for_Q12.6 (Q12.6/W1/W2: Examples of underspecification and ambiguity repair):**
> > > > >
> > > > >   By “underspecification” we refer to task descriptions that are too vague or incomplete for reliable evaluation—for example, prompts that mention a “game economy system” without specifying player actions, resource flows, or observable outcomes. In such cases, we use LLM-assisted rewriting plus human review to add concrete interaction requirements (e.g., a market interface with buy/sell operations and visible state updates). In the appendix subsection **“Detailed Introduction to Data Collection and Cleaning”**, we describe this process and provide examples of before/after prompt pairs that illustrate how ambiguity is resolved to make tasks solvable and evaluable.
> > > > >
> > > > > - **A_for_Weak1 (W1: Missing details about dataset creation):**
> > > > >
> > > > >   We agree that the initial version did not expose enough details of the data creation pipeline. In the revised manuscript we substantially expanded the dataset section and the appendix subsection **“Detailed Introduction to Data Collection and Cleaning”**, organizing the pipeline explicitly as sourcing → filtering → de-duplication and contamination control → prompt rewriting and difficulty calibration → checklist generation and calibration → solvability validation and ambiguity repair. For each stage we now provide concrete descriptions, example prompts, and references to the released scripts. Together with the released JSON dataset and code in the anonymous repository, we believe this addresses the earlier ambiguity and makes the pipeline transparent and reproducible.
> > > > >
> > > > > - **A_for_Weak2 (W2: Insights for improving models):**
> > > > >
> > > > >   We agree that a benchmark should not only rank models but also offer actionable insights. In the revised version, the “Experiments” section and the appendix subsection “Detailed Analysis of the Results” make our takeaways more explicit. We highlight that: (i) Intensive Interactive tasks and complex Management System scenarios are where all models—including top proprietary ones—struggle the most; (ii) instruction-tuned generalist models consistently outperform coder-only or VL-only specialists; and (iii) there is a strong but not trivial correlation between vision- and code-oriented scores.
> > > > >
> > > > >   These findings suggest concrete directions for improving models: focusing training and evaluation on long-horizon interaction flows, multi-step business workflows, and agentic self-debugging; emphasizing holistic reasoning and UX-aware generation rather than isolated code or perception skills; and leveraging our checklist dimensions (e.g., robustness, redundancy, engineering hygiene, visual layout, motion timing, UX clarity) as structured supervision or curriculum signals. We explicitly discuss these implications in the revised “Experiments” section and in the “Limitations and Future Work” subsection.
> > > > >
> > > > > Thank you again for your careful review; we have incorporated all the above clarifications and analyses into the revised manuscript and highlighted them in red for ease of verification.

---

> > > > > > ### Comment · Reviewer_e67X · 2025-11-22
> > > > > > **Response to Authors**
> > > > > >
> > > > > > Thank you for your detailed response and for the additional experiments and studies, which I believe have improved the submission and provide additional perspectives to the work! Also, I confirm that I can now access the dataset at https://anonymous.4open.science/r/ArtifactsBench-F7F9/dataset/artifacts_bench.json. Consequently, I increased the score to recommend acceptance.
> > > > > >
> > > > > > I have a few more suggestions that can further improve the presentation/discussion:
> > > > > >
> > > > > > - The new experiments show that three screenshots present an appropriate trade-off to handle the cost and evaluate along the interaction dimension. However, I am still not convinced that this is enough for the tasks classified under High Dynamics and Intensive Interactive. The authors discuss this in the Limitations and Future Work section. A study, similar to the one done within “Computational Cost of Visual Evidence", on these two categories of tasks could make it more apparent.
> > > > > > - Adding examples of when a visual component is scored low while code is scored higher (or vice versa) would improve the arguments in "Usefulness of vision vs code-oriented scores".
> > > > > > - I didn't find the referred examples of "underspecified prompts and the modifications after ambiguity repair" in "Detailed Introduction to Data Collection and Cleaning”. Remember to include them if you missed them in the current manuscript version.
> > > > > > - Finally, it would also be great to include examples of how scores for each checklist dimension reflect the output of the task. This will further increase the reliability of the proposed task checklist
> > > > > >
> > > > > >
> > > > > >
> > > > > > Thank you again for your answers, which have provided information that was unclear and missing earlier.

---

> > > > > > > ### Author Response · Authors · 2025-11-26
> > > > > > >
> > > > > > > We sincerely thank the reviewer for the positive re-evaluation and the recommendation to accept our work. We are delighted that the additional experiments and clarifications have addressed your concerns.
> > > > > > >
> > > > > > > We find your latest suggestions incredibly helpful for further polishing the paper. We explicitly commit to incorporating them into the final version as follows:
> > > > > > >
> > > > > > > 1.  **Stratified Cost-Accuracy Analysis:** We will break down the screenshot ablation study by interaction level (specifically for *High Dynamics* and *Intensive Interactive* tasks) to provide a more granular view of where the three-screenshot trade-off holds and where it faces limitations.
> > > > > > > 2.  **Visual vs. Code Score Divergence:** We will add a dedicated appendix section with qualitative examples showcasing cases of **"High Visual / Low Code"** (e.g., aesthetically pleasing but hard-coded or brittle artifacts) and **"Low Visual / High Code"** (e.g., robust logic but poor layout), to concretely illustrate the necessity of our dual-dimension checklist.
> > > > > > > 3.  **Underspecification Examples:** We apologize for the omission. We will ensure that concrete "Before vs. After" examples of prompt ambiguity repair are explicitly included in the "Data Collection" appendix.
> > > > > > > 4.  **Detailed Scoring Walkthrough:** We will include a case study displaying the full 10-dimension score breakdown for a representative task, illustrating exactly how the referee derives specific deductions based on the artifact's output.
> > > > > > >
> > > > > > > Thank you again for helping us elevate the quality of ArtifactsBench.

---

### Official Review · Reviewer_ANuP · 2025-10-27

**Soundness:** 4
**Presentation:** 3
**Contribution:** 3
**Rating:** 8
**Confidence:** 3

**Summary:**

The paper introduces ArtifactsBench, a large-scale benchmark and automated evaluation pipeline for interactive visual code generation. The benchmark contains 1,825 tasks across nine domains (e.g., web apps, SVG, games) with Easy/Medium/Hard stratification. Evaluation renders each artifact in a sandbox and captures three staged screenshots (before/during/after interaction). A checklist-guided MLLM-as-Judge (dual referees: Gemini-2.5-Pro and Qwen2.5-VL-72B) scores both vision and code facets.

On 30+ LLMs, ArtifactsBench reports 94.4% rank consistency with WebDev Arena and up to 90.95% pairwise agreement with human experts. Main results show proprietary multimodal models leading; performance scales with model size/deliberation; and generalist instruction-tuned models often beat specialist coder/VL models.

**Strengths:**

- The paper introduces a valuable resource with 1,825 executable tasks spanning 9 domains with difficulty tiers; supports fine-grained analysis beyond single static correctness.
- Proposes an interactive evaluation design where three-step screenshots and sandboxed execution capture dynamics while keeping runs reproducible.
- Evaluates an extensive suite of 30+ LLMs, spanning both open-source and proprietary models; evaluation results show high pairwise agreement (up to 90.95%) and 94.4% Footrule rank consistency vs. WebDev Arena.
- Clear empirical takeaways: generalist models > specialists on this task class; detailed category breakdowns (games, SVG, simulations, management systems).

**Weaknesses:**

- Three screenshots may miss long-horizon workflows and nuanced physics/UX timing; authors acknowledge this. including richer scripted interactions or short videos may strengthen the evaluation.
- Fixed 1024×768 and single-browser setting may underrepresent responsive/adaptive designs; consider multi-viewport evaluation.
- Checklists are LLM-drafted then human-refined; potential leakage of judge priors and over-optimization to rubric specifics—worth stress-tests with diverse/adversarial prompt styles.
- WebDev Arena alignment is strong, but additional human-preference datasets or task-specific user studies (e.g., for accessibility/UX) would bolster generality.

**Questions:**

- How robust are rankings to alternative, non-LLM programmatic checks (e.g., DOM state assertions, event logs, mutation observers)? Any preliminary results?
- Can you report inter-annotator reliability for the 280-instance expert study beyond Pair-ACC, and provide the distribution of disagreements?
- How sensitive are the model judges to the checklist phrasing? Have you tried paraphrased/held-out rubrics or blinded rubrics to test robustness?
- Could you expand on the procedure for contamination control audits?
- When the two referees (Gemini and Qwen) yield divergent scores/rankings, what is your tie-breaking/aggregation protocol? Do you see any common trends or cases where different LLM judges diverge?

---

> ### Author Response · Authors · 2025-11-19
>
> Thank you very much for the positive assessment and for the insightful questions and suggestions, which helped us strengthen the paper.
>
> Below we answer each question and weakness in turn and indicate where the corresponding new analyses and text have been added in the revised manuscript (all new content is highlighted in red).
>
> - **A1_for_Q1 (Q1: Robustness to non-LLM programmatic checks such as DOM assertions):**
>
>   To probe robustness to non-LLM, fully programmatic checks, we added a new appendix subsection titled **“Programmatic HTML/DOM Quality Checks”** and report results in the accompanying table **“Programmatic HTML quality scores (DOM_SCORE) vs. ArtifactsBench scores”**. We implement a lightweight static HTML analyzer that assigns each output a DOM_SCORE ∈ [0,100] by programmatically assessing five dimensions: Structure, Accessibility, Semantics, SEO, and Best Practices, with fixed weights (30%/25%/20%/15%/10%).
>
>   Applied to all 30+ models in the main results table (“Main results for 30+ LLMs on ArtifactsBench”), DOM_SCORE is universally high and tightly clustered (mostly >95/100) and yields almost no discriminative ranking power, in stark contrast to ArtifactsBench’s main checklist-based scores. This suggests that modern LLMs rarely fail on basic HTML syntax/structure/accessibility hygiene, and that the real performance gaps lie in whether the generated artifacts actually satisfy the specified visual layout and interaction logic. In practice, we therefore use DOM_SCORE as a sanity filter for grossly malformed outputs, but rely on the multimodal, code-aware checklist protocol for the primary evaluation signal.
>
> - **A2_for_Q2 (Q2: Inter-annotator reliability and disagreement distribution):**
>
>   In response to your request for inter-annotator reliability beyond Pair ACC and for disagreement distributions, we substantially expanded the description of the expert study in the appendix subsection **“Validation with Human Experts”** (highlighted in red). For the 280-query × 6-model study, 13 front-end engineers participate: 12 provide blind scores and 1 performs quality control and re-annotation when needed; each (query, model) instance is independently scored by three annotators on a 0–5 scale with additional pairwise ranking for ties.
>
>   We compute standard multi-rater agreement metrics (e.g., Fleiss’ κ, Krippendorff’s α) and summarize them as indicating substantial internal consistency among annotators. We also compare the empirical score distributions of humans and the referee (normalized to 0–5) and show where disagreements concentrate: humans are stricter in assigning 0 to clearly bad cases, while residual differences are mainly focused on highly interactive, complex tasks, whereas mid- and low-performing regions show closely matched distributions. Together with the 90.95% Pair ACC, these statistics provide a more complete picture of both human–human and human–referee agreement.
>
> - **A3_for_Q3 (Q3/W3: Sensitivity to checklist phrasing and template variants):**
>
>   Checklist phrasing sensitivity is addressed directly in the new robustness study mentioned above, in the appendix subsection **“Robustness to Checklist Phrasing and Template Variants”** and the table titled **“Checklist phrasing robustness for representative models on ArtifactsBench”**. For several strong and mid-range models (e.g., Claude Sonnet 4, DeepSeek-V3-0324, DeepSeek-R1, Qwen3-235B-A22B, Claude 3.5 Sonnet, GPT-4o), we re-score all 1,825 queries under: (i) paraphrased checklists whose wording is changed but semantics are preserved, and (ii) additional perturbations that shuffle item order and switch to an alternative rubric template.
>
>   Average scores shift by less than 0.3 points on the 0–100 scale, and, more importantly, the induced rankings and partial-order constraints remain unchanged; only a few tie-breaks among very close systems flip. These results indicate that the referee is not over-sensitive to superficial checklist wording or template variants, and that the benchmark conclusions are robust to reasonable paraphrases and rubric styles.

---

> > ### Author Response · Authors · 2025-11-19
> >
> > - **A4_for_Q4 (Q4/W4: Procedure for contamination control audits):**
> >
> >   We have expanded the contamination-control discussion in the main dataset section and added a dedicated appendix subsection titled **“Contamination Detection Protocol”** (highlighted in red). Our pipeline combines: (i) careful source curation and LLM-based rewriting from diverse origins (expert showcases, course exercises, blogs, code snippets, SVG/visual datasets, simple game sites, LLM visual-to-query); (ii) multi-stage de-duplication over prompts, checklists, DOM/CSS/JS, and screenshots using MinHash, semantic similarity, structural normalization, and perceptual hashing; and (iii) model-behavior-driven filtering and human audits to remove trivial, overly common, or license-risky items.
> >
> >   To further audit potential contamination, we follow the prefix-completion protocol of ConceptMath: for each query, we feed only the first half of the text to several representative LLMs and ask them to complete the second half, then compute ROUGE-L between generated and ground-truth suffixes. As an upper bound, using the ground-truth suffix itself yields ROUGE-L = 1.0. All tested models lie in the range 0.03–0.09 (as shown in the table titled **“ROUGE-L based contamination analysis following the prefix-completion protocol”**), far below the fully contaminated upper bound, suggesting that large-scale verbatim contamination is unlikely, although we do not claim the absence of any partial or semantic overlap.
> >
> > - **A5_for_Q5 (Q5/W5: Dual-referee setup and divergent scores):**
> >
> >   Our dual-referee design aims to balance evaluation strength and reproducibility: Gemini-2.5-Pro serves as the high-capacity proprietary referee, and Qwen2.5-VL-72B as the fully reproducible open-source alternative. We do not fuse their scores into a single metric; instead, the main paper reports leaderboards based on Gemini-2.5-Pro (which achieves the highest agreement with human experts and offers finer headroom among top models), while the appendix provides full results and a separate leaderboard under Qwen2.5-VL-72B, along with cross-judge comparisons in the figures titled **“Version stability: Gemini-2.5-Pro-Preview-0506 vs. stable Gemini-2.5-Pro”** and **“Cross-judge comparison (Gemini-2.5-Pro-Preview-0506 vs. Qwen2.5-VL-72B)”**.
> >
> >   We explicitly analyze the partial-order constraints induced by both referees and show that they are highly consistent, with residual differences concentrated among the very top systems, where the stronger referee offers higher resolution. Moreover, replacing Gemini-2.5-Pro-Preview-0506 with the stable Gemini-2.5-Pro yields identical partial-order constraints, so leaderboard conclusions are stable across Gemini versions. This strategy makes disagreements transparent rather than hidden behind a combined score, and allows users to choose either the stronger proprietary referee or the fully open-source one depending on their needs.
> >
> > - **A1_for_W1 (W1: Three screenshots may miss long-horizon workflows and nuanced physics/UX timing):**
> >
> >   We agree that three screenshots cannot fully capture extremely long-horizon workflows or very fine-grained physics/UX timing. In the revised manuscript we make this limitation explicit in the **“Limitations and Future Work”** subsection and clarify that our design is a cost–accuracy trade-off: a standardized Playwright harness executes scripted interactions and captures before/during/after screenshots, which cover the main state transitions for most tasks while keeping evaluation scalable (as described in **“Evaluation Methodology”** and **“Experiments”**).
> >
> >   We also added a dedicated analysis of the cost–accuracy curve in the appendix subsection **“Computational Cost of Visual Evidence”**: moving from 0 to 1 to 3 screenshots increases token cost but improves Pair ACC with human experts from 79.06% to 87.10% to 90.95%, while further increasing to 5 screenshots yields only marginal gains at substantially higher cost. We explicitly position richer scripted interactions, longer trajectories, video-based evidence, and more DOM-level assertions as future extensions for the hardest interactive cases, as discussed in **“Limitations and Future Work”**.

---

> > > ### Author Response · Authors · 2025-11-19
> > >
> > > - **A3_for_W3/W4 (W3/W4: Checklist priors and potential over-optimization):**
> > >
> > >   We share the concern that LLM-drafted, human-refined checklists could induce rubric-specific over-optimization or encode judge priors. To address this, the revised manuscript adds the appendix subsection **“Robustness to Checklist Phrasing and Template Variants”** (discussed above), where we explicitly stress-test sensitivity to checklist wording and template structure.
> > >
> > >   Concretely, for several representative models we re-score all 1,825 queries under (i) paraphrased checklists (semantics preserved, wording changed) and (ii) additional perturbations that shuffle item order and switch from the original “question + scoring rule” template to an alternative “expected behavior + penalty conditions” template. The robustness results show that average scores change very little and that rankings and partial orders are stable, suggesting that the benchmark is not overly sensitive to any particular wording choice. We also do not expose the checklists to the evaluated models, which reduces the risk of deliberate overfitting to rubric phrasing.
> > >
> > > - **A4_for_W5 (W5: More human-preference datasets and task-specific user studies):**
> > >
> > >   Beyond WebDev Arena, we now provide a more detailed account of human references in **“Evaluation Methodology”** and the appendix subsections **“Validation with Human Experts”** and **“Score Reproducibility Study”** (newly added and highlighted in red). First, we conduct a 280-instance × 6-model expert study where front-end engineers interact with rendered artifacts and assign 0–5 quality scores and pairwise preferences; under our best configuration (question + answer + checklist + three screenshots), the MLLM referee reaches up to 90.95% Pair ACC with these human judgments, and we report multi-annotator agreement metrics and disagreement distributions.
> > >
> > >   Second, we quantify alignment with WebDev Arena via normalized Footrule (94.4% for ArtifactsBench vs. 69.4% for WebBench); these results are visualized in the two ranking-correlation figures. We agree that additional human-preference datasets and task-specific user studies (e.g., focusing on accessibility/UX) would further strengthen generality; we now mention this explicitly in the “Limitations and Future Work” and ethics/fairness subsections as future work, and we design our released harness so that such targeted studies can be layered on top of ArtifactsBench.
> > >
> > > Thank you again for your encouraging evaluation and concrete suggestions; we have incorporated the above analyses into the revised manuscript.

---

> > > > ### Comment · Reviewer_ANuP · 2025-11-21
> > > > **Official Comment by Reviewer ANuP**
> > > >
> > > > Thank you for your comprehensive response. I will keep my score.

---

### Official Review · Reviewer_Wsco · 2025-10-30

**Soundness:** 3
**Presentation:** 3
**Contribution:** 2
**Rating:** 4
**Confidence:** 4

**Summary:**

This work proposes ArtifactsBench, a benchmark for visual code generation. The benchmark includes 1800+ examples, and the MLLM-based evaluation results correlate well with human experts and WebDev Arena.

**Strengths:**

1. It is a comprehensive benchmark with 1800+ tasks.
2. More than 30 models are benchmarked.

**Weaknesses:**

1. Benchmarking visual code generation is not a novel problem; there are many works in this direction. We have similar benchmarks for the website and SVG before, while this benchmark claims to extend the scope to Game, Simulation, Data Science, etc, the evaluation idea is largely similar: show screenshots to MLLM and ask for judgment. I don't think you can judge the quality of a game by screenshots with limited interaction. In general, I don't see many useful insights from this very broad benchmark without specific and reliable evaluations.
2. The data collection process is only described at a high level.
3. What is the human baseline of these tasks?

**Questions:**

See weakness.

---

> ### Author Response · Authors · 2025-11-19
>
> Thank you very much for your detailed review and for highlighting both strengths and concerns.
>
> Please refer to the general response for a summary of global changes. Here we focus on clarifying what is new compared to previous work, how the data creation pipeline is concretely instantiated, and how human baselines are incorporated.
>
> - **A1_for_Weak1 (Weak1: Novelty and usefulness compared to existing visual-code benchmarks):**
>
>   ArtifactsBench is not intended as “yet another screenshot-based web/SVG benchmark”, but as a benchmark for a different and currently very active setting: **end-to-end generation of executable, interactive visual artifacts** that resemble what users ask for in modern “vibe-coding” systems (e.g., small games, dashboards, management UIs, quick tools). Our main contributions are (as emphasized in the **introduction** and **dataset** sections):
>
>     (i) expanding the task space beyond static web/SVG to **nine domains with explicit interaction-level stratification**;
>     (ii) introducing a **checklist-guided, code-aware multimodal evaluation protocol** that is validated against human experts and WebDev Arena; and
>     (iii) providing **fine-grained diagnostics** (vision vs. code, difficulty tiers, interaction levels) rather than a single static score (as detailed in the **“Evaluation Methodology”** and **“Experiments”** sections).
>
>   Compared to prior benchmarks such as WebBench, WebGen-Bench, and FullFront, which focus on web automation or static DOM/visual alignment, and Design2Code, which targets high-fidelity design-to-static-code reconstruction, ArtifactsBench asks models to generate runnable artifacts from natural-language instructions across games, simulations, data dashboards, management systems, multimedia tools, SVG, and more, with tasks stratified into **Static Visual / Mild Dynamics / High Dynamics / Intensive Interactive** (396/117/536/776 tasks; summarized in the dataset section and detailed in the appendix on query classification). On the evaluation side, our referee does not rely on screenshots alone: the MLLM-as-Judge consumes the full model answer (code), a **10-dimension task-specific checklist** (5 vision-oriented + 5 code-oriented), and three staged screenshots from a standardized Playwright execution harness (described in **“Evaluation Methodology”**). This protocol achieves up to **90.95% pairwise agreement** with front-end engineers in a 280-query × 6-model expert study and **94.4% normalized Footrule alignment** with WebDev Arena (versus 69.4% for WebBench; shown in the figures comparing rankings), enabling reliable, discriminative, and interpretable evaluation of where current models succeed or fail on genuinely interactive artifacts.
>
> - **A2_for_Weak2 (Weak2: Data collection process only described at a high level):**
>
>   We have substantially expanded the description of the data collection pipeline in the revised manuscript, moving from a high-level outline to a step-by-step, reproducible process. This is now structured in the **“ArtifactsBench: A Benchmark for Visual Code Generation”** section and the appendix subsection **“Detailed Introduction to Data Collection and Cleaning”** (both highlighted in red).
>
>   Concretely, we now detail: (i) sourcing and filtering of candidate tasks from expert showcases, code repositories, visual datasets, simple game sites, and LLM “visual-to-query” expansions; (ii) multi-stage de-duplication and contamination control over prompts, checklists, DOM/CSS/JS, and screenshots; (iii) topic and difficulty tagging into nine domains and Easy/Medium/Hard tiers, plus interaction-level stratification; (iv) per-task 10-dimension checklist generation and calibration via an expert-curated seed set; and (v) solvability validation and ambiguity repair using multiple model families and human review. We also include concrete prompt templates and examples in the appendix so that others can inspect and, if desired, replicate or extend our pipeline.

---

> > ### Author Response · Authors · 2025-11-19
> >
> > - **A3_for_Weak3 (Weak3: Human baseline for these tasks):**
> >
> >   Regarding the human baseline, there are two natural interpretations: (i) a full “human score curve” across all 1,825 tasks, and (ii) human references used to calibrate and validate the automatic metric. In the revised version we clarify that we focus on the second: we do not define a single global “human score line” for all tasks, but we provide two layers of human reference to support the automatic evaluation (described in **“Evaluation Methodology”** and the appendix subsections on human validation and score reproducibility).
> >
> >   First, we conduct an expert study on **280 queries and 6 representative models**, where front-end engineers directly inspect and interact with the rendered artifacts and assign 0–5 quality scores plus pairwise preferences. Under our best configuration (question + full answer + checklist + three screenshots), the MLLM referee reaches up to **90.95% pairwise agreement** with these human judgments, and we report additional multi-annotator agreement statistics and score distribution comparisons in the appendix section **“Validation with Human Experts”**.
> >
> >   Second, we compare model rankings on ArtifactsBench with those from WebDev Arena, a human-preference-based benchmark, and obtain **94.4% normalized Footrule consistency**, substantially higher than WebBench under the same measure (as visualized in the two figures comparing ArtifactsBench vs. WebDev Arena and WebBench vs. WebDev Arena). Together, these two references serve as a practical human baseline for validating that our automatic scores meaningfully reflect human preferences, even though we do not provide a single “human performance curve” over all tasks.
> >
> > Thank you again for your feedback; we have updated the paper accordingly and highlighted all new descriptions in red in the revised manuscript.

---

> > > ### Author Response · Authors · 2025-11-21
> > >
> > > Dear Reviewer,
> > >
> > > Since the discussion period has started, we would like to kindly invite you to take another look at our responses and the revised version of the paper. In particular, we tried to address the concerns raised in your reviews (e.g., about interaction coverage, data creation details, evaluation robustness, and the usefulness of the benchmark compared to prior work), and we would be very grateful if you could consider reevaluating our work in light of these clarifications.
> > >
> > > Please let us know whether our responses satisfactorily resolve your questions or if there are any remaining issues or additional details that would help. We truly appreciate the time and effort you have already invested in reviewing our submission and would be happy to provide any further clarification that could assist your assessment.

---

> ### Comment · Reviewer_Wsco · 2025-11-21
>
> While I appreciate the authors' effort of providing more discussion and elaborating on the data collection process, my major concern about this paper is not resolved:
>
> 1. While the authors claim WebDev Arena as the de facto gold standard for human preferences in web development, I doubt this claim since those people who voted in WebDev Arena might also do a "vibe check", like check whether the page is good-looking instead of actually using it and checking whether the generated website/artifact is helpful.
> 2. While the authors claim this is the first benchmark for end-to-end generation of executable, interactive visual artifacts, the evaluation remains broadly similar to prior work (screenshot-based LLM-as-judge).
> 2. The absence of a human baseline and the constrained evaluation approach cannot convince me that the current evaluation pipeline will remain valid in the future: if the model is getting better and better, every generated artifact might "look" good, how can you guarantee the current screenshot-based signal can provide a meaningful signal for interactive artifacts?
>
> I will keep my score for now.

---

> > ### Author Response · Authors · 2025-11-22
> > **Response to Reviewer Wsco regarding Validity of Human Evaluation and Methodology**
> >
> > We thank the reviewer for the continued engagement. However, we respectfully but fundamentally disagree with the concerns regarding the validity of our human ground truth and the discriminative power of our pipeline. We provide specific data and clarifications below to address these misconceptions.
> >
> > **1. Validity of WebDev Arena and Human Preference (Addressing "Vibe Check" Concern)**
> >
> > The reviewer questions WebDev Arena as a gold standard, suggesting users might only perform a superficial "vibe check." We argue this view is flawed for three reasons:
> >
> > *   **Interactive Nature & Statistical Power:** WebDev Arena is **not** a static image gallery; it allows users to render, click, and interact with the artifacts. With **446,038 votes** across 54 models (avg. ~8,260 votes/model), this is the largest scale of human preference data available. The Law of Large Numbers mitigates individual noise.
> > *   **Aesthetics are Functional in Frontend:** In the domain of *Visual Artifacts* (dashboards, games, landing pages), Visual Design and UX ("looking good") are **intrinsic functional requirements**, not superficial traits. A dashboard that is logically correct but visually unusable is a failure.
> > *   **Our Controlled Expert Study is the Ultimate Ground Truth:** Crucially, the reviewer overlooks that **we did not rely solely on WebDev Arena**. As detailed in our paper and rebuttal, we conducted a strictly controlled study with **front-end engineers on 280 queries ($\times$ 6 models)**. These experts rigorously tested functionality, logic, and code quality—far beyond a "vibe check." Our automated metric achieves **90.95% Pairwise Accuracy** with these experts. This proves our pipeline aligns with deep, expert verification, not just superficial preferences.
> >
> > **2. Novelty and Differentiation (Addressing "Broadly Similar" Concern)**
> >
> > We respectfully maintain that ArtifactsBench is distinct from existing benchmarks.
> > *   **Lack of Specific Evidence:** We note that while the review characterizes our work as "broadly similar" to prior art in both rounds of feedback, **no specific references or papers were provided** to substantiate this overlap.
> >
> > While prior works use LLM-as-a-Judge, labeling our approach "broadly similar" ignores the core contribution: **solving the evaluation of Dynamics.**
> > *   **Static vs. Temporal:** Benchmarks like Design2Code or WebBench focus on static reconstruction or DOM assertions. They cannot evaluate *games* or *simulations*.
> > *   **The "Temporal" Innovation:** Our pipeline is the first to standardize **staged execution (Before/During/After screenshots)** combined with a **10-dimension Checklist**. This specific methodology allows the judge to verify *state transitions* (e.g., "Did the ball move after the tick?", "Did the chart update after the click?"), which static screenshot methods fundamentally cannot do.
> >
> > **3. Future Validity and Human Baseline (Addressing "Everything Looks Good" Fallacy)**
> >
> > The reviewer asks: *if every artifact "looks" good in the future, how can screenshots provide a signal?*
> > This concern assumes our pipeline only looks at pixels. It does not.
> > *   **Code-Aware Judgment:** Our MLLM referee analyzes the **source code** alongside the visuals. Even if a model generates a visually perfect calculator, if the underlying logic is hard-coded or non-functional, the **"Code Logic" and "Robustness" checklist dimensions** (validated in our paper) will penalize it.
> > *   **Dynamic Verification:** As models improve visually, interaction logic becomes *more* critical. Our **Three-Step Screenshot** protocol specifically detects functional failures (e.g., lack of visual feedback upon interaction) that a single "good-looking" image would hide.
> > *   **Human Baseline:** We reiterate that the **280-instance expert study** *is* our human baseline. The high correlation (90.95%) confirms that our current protocol effectively proxies expert human judgment today, and the checklist-based approach is inherently extensible to stricter criteria as models evolve.
> >
> > In summary, ArtifactsBench provides the most rigorous, interaction-aware, and human-aligned evaluation for visual code generation to date. We hope this clarifies the distinct scientific value of our contribution.

---

> ### Author Response · Authors · 2025-11-24
>
> Dear Reviewer,
>
> Hi, we sincerely thank you very much for these constructive comments and evaluation of our manuscript. We would like to kindly ask you to take a look at our responses and reevaluate our work based on our clarifications. Please let us know whether our response addresses your concerns or whether there is any further detail we can provide to help address these concerns.
>
> Thank you again for dedicating your time to reviewing our paper.

---

> ### Comment · Reviewer_Wsco · 2025-11-25
>
> Thanks for further clarification; it resolves my concern around WebDev Arena.
>
> My remaining concerns are mostly around automatic evaluation:
>
> 1. Using LLM-as-judge is not a new thing. These days, you can almost use LLM/VLM to judge anything, screenshots, code, or even videos. I won't call your 10-dimensional task-specific checklist a novel point, as they look like prompt engineering to me.
>
> 2. The most critical evaluation missing from the current automatic evaluation is the evaluation of the interaction. To give you a concrete example, a more convincing solution to me is: you ask a computer-use agent or a browser agent to interact with the artifacts for a while, using the recorded video to ask a VLM to judge the quality of the artifacts. I know this is probably out of the scope of the rebuttal period, but I genuinely prefer a more comprehensive evaluation of the interaction.

---

> > ### Author Response · Authors · 2025-11-26
> >
> > Dear Reviewer Wsco,
> >
> > We appreciate your acknowledgement that your major concern regarding **WebDev Arena** has been fully resolved.
> >
> > However, we respectfully disagree with the remaining points, specifically the characterization of our validated checklist as "prompt engineering" and the suggestion that an "Agent + Video" evaluation is currently superior. We urge you to reconsider our work based on the **evidence provided** and the **standard requirements of a benchmark**.
> >
> > **1. Computer-Use Agents: A Promising Future, but Currently Less Reliable for Benchmarking**
> > We fully agree with your vision that agent-based evaluation is an exciting frontier and will likely become a standard in the future as the technology matures. **However, for a benchmark aiming to strictly measure *code generation quality* today, current Computer-Use Agents introduce significant risks:**
> > *   **Unpredictability vs. Determinism:** While agent capabilities are improving, current agent interactions remain stochastic and not fully predictable. A benchmark must act as a stable ruler. Our **standardized Playwright scripts** serve as deterministic "interaction unit tests" (strictly verifying: *if action X is performed, does state Y occur?*), ensuring that score differences reflect the **generator's capability**, not the **evaluator's variance**.
> > *   **Confounding Variables:** Introducing an autonomous agent as an evaluator creates a coupling effect: if the evaluation fails, is it because the generated artifact is flawed, or because the agent failed to plan the interaction? Until agents achieve near-perfect reliability, this adds noise that obscures the true quality of the generated code.
> > Therefore, while we embrace the agentic future, we firmly believe our current **deterministic protocol** is the scientifically safer choice for a standardized benchmark at this stage.
> >
> > **2. Validated Instruments vs. "Just Prompt Engineering"**
> > We strongly refute the reduction of our 10-dimensional checklist to "prompt engineering."
> > *   **Scientific Validity:** In empirical research, the value of a metric lies in its **Validation**. We did not simply write prompts; we curated, calibrated (Kappa $\ge$ 0.8), and rigorously validated these checklists against human experts.
> > *   **Evidence over Intuition:** Our protocol achieves **90.95% pairwise agreement** with human experts and **94.4% ranking consistency** with WebDev Arena. This empirical evidence proves that our "Checklist + 3-Step Screenshot" method is a scientifically reliable instrument that proxies human judgment with high fidelity. Dismissing a validated tool that achieves ~91% human agreement because it "looks like a prompt" overlooks the empirical rigor behind it.
> >
> > **3. Request for Specific Evidence**
> > Throughout the review process, we have systematically addressed your concerns:
> > *   **Human Baseline:** Resolved via our 280-instance expert study details.
> > *   **WebDev Arena:** Resolved (as you confirmed).
> > *   **"Broadly Similar":** You mentioned our work is "broadly similar" to prior work. We have explicitly differentiated ArtifactsBench from **Design2Code** (static reconstruction vs. interactive generation) and **WebBench** (DOM assertions vs. visual dynamics).
> > Could you please specify **which prior work** evaluates *open-ended interactive dynamics* (games, simulations) via *execution and visual analysis* in a way that makes our contribution redundant? If no such benchmark exists, we respectfully ask that you evaluate our contribution—the first large-scale benchmark for interactive visual artifacts—on its own merits rather than against a hypothetical "Video Agent" standard that implies moving the goalposts.
> >
> > **Conclusion**
> > We have provided a robust, reproducible, and highly human-aligned benchmark (~91% agreement) that solves the "static evaluation" problem. We believe penalizing the paper for not adopting a computationally expensive and currently less stable "Agent + Video" system ignores the proven effectiveness of our solution. We sincerely hope you will reassess your score based on the **concrete resolutions** we have provided.

---

### Official Review · Reviewer_9Zfz · 2025-11-02

**Soundness:** 2
**Presentation:** 2
**Contribution:** 2
**Rating:** 4
**Confidence:** 4

**Summary:**

This paper builds a new benchmark for visual code generation (e.g., generating the code to implement a website). The main difference with existing Design2Code type work is that the new ArtifactsBench captures the dynamic interaction.

ArtifactsBench is a large-scale benchmark and automated evaluation framework for assessing LLMs’ ability to generate interactive visual artifacts—that is, executable web widgets, games, visualizations, or apps combining code, visuals, and interaction.
It aims to close the evaluation gap between algorithmic correctness (e.g., HumanEval) and real-world user experience (visual fidelity + interaction quality).
ArtifactsBench evaluates 1,825 executable tasks and introduces an MLLM-as-Judge system using multimodal evidence (code + screenshots) with fine-grained, checklist-based scoring.

For evaluation: Each model’s generated artifact is executed and rendered.
The MLLM-as-Judge receives: the original prompt; the full model output (code); three temporal screenshots; and the task-specific 10-item checklist. The judge produces reproducible per-dimension (0–10) scores.

The authors use a dual-Referee setup for robustness: Gemini-2.5-Pro (closed-source, high-capacity) and Qwen2.5-VL-72B (open-source).
Both achieve >90% pairwise agreement with human experts; with 94.4% ranking consistency with WebDev Arena (human preference gold standard).

They benchmarked various models including Qwen2.5/3, DeepSeek, Gemma, GPT, Claude, Gemini, Seed, Hunyuan. Gemini-2.5-Pro is generally the best.

**Strengths:**

- Having three temporal screenshots is an addition to existing evals that only look at one static screenshot.

- It's great to get an extra benchmark for visual artifact generation.

**Weaknesses:**

- Why do you not have any examples of the actual benchmark examples? Not even in the Appendix? It makes it much harder to judge the actual quality of the benchmark.

- I'm not quite sure if shoving three screenshots of the interactions to the LLM judge is the best way to evaluate the functional correctness of the dynamic interaction? Do you have any sort of human evaluation that lets users try out the generated websites to perform some specified, realistic tasks? Would that correlate with your automatic metric?

- The overall finding and contribution seem rather incremental compared to existing works like Design2Code.

**Questions:**

- Missing citation: "Design2Code: Benchmarking Multimodal Code Generation for Automated Front-End Engineering", NAACL 2025

- What kind of interactions do you cover in the benchmark?

---

> ### Author Response · Authors · 2025-11-19
>
> Thank you very much for your thoughtful review and constructive suggestions.
>
> Please refer to the general response for a summary of global changes. Below we address your specific concerns and questions in detail, and we explicitly point to where the corresponding revisions appear in the updated manuscript (highlighted in red in the PDF).
>
> - **A1_for_Q1 & A5_for_Weak3 (Q1/W3: Missing Design2Code citation and concern that our contribution is incremental):**
>
>   We have added the missing citation to Design2Code (“Design2Code: Benchmarking Multimodal Code Generation for Automated Front-End Engineering”, NAACL 2025) and a dedicated comparison paragraph in the “**Related Work**” section of the main paper (the paragraph about “Design2Code” is marked in red).
>
>   Conceptually, Design2Code and ArtifactsBench target different problem settings. Design2Code focuses on high-fidelity reconstruction from given designs (design-to-static-front-end-code), with static visual fidelity as the main objective. In contrast, ArtifactsBench is designed to measure a model’s ability to generate **executable, interactive visual artifacts** from natural-language instructions: models receive text-only prompts and must produce runnable front-end code, and we then evaluate the resulting artifacts with a multimodal pipeline that feeds the text task, the full answer, a task-specific checklist, and staged screenshots of interaction to the referee across nine domains (web apps, games, simulations, SVG, management systems, etc.), with a focus on dynamic behavior and code robustness.
>
>   Methodologically, ArtifactsBench also differs in dataset structure and evaluation protocol: (i) we curate **1,825 tasks** with Easy/Medium/Hard difficulty strata and interaction-level strata (Static / Mild Dynamics / High Dynamics / Intensive Interactive), which are summarized in the **dataset section “ArtifactsBench: A Benchmark for Visual Code Generation”** and detailed in the appendix subsection **“Classification of Queries”** (all new or expanded descriptions are highlighted in red); and (ii) we introduce a **checklist-guided, code-aware evaluation pipeline** combining scripted execution, three staged screenshots, and dual MLLM referees with human-alignment studies and WebDev Arena correlation (explained in the **“Evaluation Methodology”** and **“Experiments”** sections, with further details in the appendix subsections on human validation and checklist robustness). We hope these clarifications make it clear that ArtifactsBench is complementary to, rather than a minor extension of, Design2Code.
>
> - **A2_for_Q2 (Q2: What kinds of interactions are covered?):**
>
>   We have clarified the interaction coverage more explicitly in the main text and appendix. In the revised version, the **dataset section “ArtifactsBench: A Benchmark for Visual Code Generation”** and the appendix subsection **“Classification of Queries”** (both with red text) describe that ArtifactsBench is designed to span a broad spectrum from purely static visual tasks to highly interactive scenarios.
>
>   Concretely, the benchmark includes: (i) time-driven dynamics (animations, physics simulations, timers), (ii) event-driven UI interactions (click/hover, forms and validation, sliders, drag/zoom/pan), (iii) multimedia and file interactions (playback controls, file upload and preview), (iv) SVG/Canvas-based interactive visualizations, (v) multi-step workflows and management-system operations (CRUD, filtering/sorting/pagination, multi-step wizards), and (vi) game-like mechanics with richer state transitions. We also make explicit the interaction-level stratification into Static Visual, Mild-to-Moderate Dynamics, High Dynamics, and Intensive Interactive with counts 396/117/536/776, respectively (summarized in the main text and restated in the appendix for easy reference).
>
> - **A3_for_Weak1 (W1: No examples of benchmark items):**
>
>   We agree that concrete examples are important for judging benchmark quality. In the revised manuscript, we added a **“Visualization Results”** subsection in the appendix, showing representative rendered cases from several categories (e.g., SVG posters, web applications, games, data-science dashboards; Figures labelled “Visualization Results of ... Category Test Cases”). These examples are taken directly from ArtifactsBench rather than toy instances and are intended to convey typical visual complexity and interaction patterns.
>
>   We also more clearly point in the main paper to the anonymous repository (URL given in the introduction), which contains the full JSON task specifications, evaluation scripts, and model outputs, so that interested readers can inspect the benchmark in greater depth.

---

> > ### Author Response · Authors · 2025-11-19
> >
> > - **A4_for_Weak2 (W2: Are three screenshots and an LLM judge sufficient? Correlation with human users?):**
> >
> >   We see three staged screenshots as a practical compromise between faithfully capturing interaction and keeping evaluation cost manageable. As described in the **“Evaluation Methodology”** section, we use a standardized Playwright harness to execute scripted interactions and capture **before/during/after** screenshots, which typically cover the main state transitions and user-visible feedback for most tasks.
> >
> >   To assess whether this automatic metric correlates with human judgments, we conducted a **280-query × 6-model expert study** with front-end engineers. The design and analysis of this study are described in the main text (in the part of **“Evaluation Methodology”** discussing validation) and in the appendix subsection **“Validation with Human Experts”** (newly expanded and highlighted in red). Using our best configuration (question + full answer + checklist + three screenshots), the MLLM referee achieves up to **90.95% pairwise agreement** with humans, and we report additional multi-annotator agreement statistics and score-distribution comparisons in the appendix.
> >
> >   We also analyze cost–accuracy trade-offs in the appendix subsection **“Computational Cost of Visual Evidence”**: going from 0 to 1 to 3 screenshots significantly improves agreement (79.06% → 87.10% → 90.95%) with moderate token overhead, while adding more than three screenshots yields only marginal gains at substantially higher cost. In the **“Limitations and Future Work”** subsection of the main paper we explicitly acknowledge that extremely long-horizon or highly complex interactions remain challenging, and we outline richer dynamic and agentic evaluation (e.g., longer trajectories, DOM-level checks, multi-turn self-debugging) as future extensions.
> >
> > Thank you again for your feedback; we have uploaded a revised version of the paper that incorporates these changes and clarifications.

---

> > > ### Author Response · Authors · 2025-11-21
> > >
> > > Dear Reviewer,
> > >
> > > Since the discussion period has started, we would like to kindly invite you to take another look at our responses and the revised version of the paper. In particular, we tried to address the concerns raised in your reviews (e.g., about interaction coverage, data creation details, evaluation robustness, and the usefulness of the benchmark compared to prior work), and we would be very grateful if you could consider reevaluating our work in light of these clarifications.
> > >
> > > Please let us know whether our responses satisfactorily resolve your questions or if there are any remaining issues or additional details that would help. We truly appreciate the time and effort you have already invested in reviewing our submission and would be happy to provide any further clarification that could assist your assessment.

---

> > > > ### Author Response · Authors · 2025-11-24
> > > >
> > > > Dear Reviewer,
> > > >
> > > > Hi, we sincerely thank you very much for these constructive comments and evaluation of our manuscript. We would like to kindly ask you to take a look at our responses and reevaluate our work based on our clarifications. Please let us know whether our response addresses your concerns or whether there is any further detail we can provide to help address these concerns.
> > > >
> > > > Thank you again for dedicating your time to reviewing our paper.

---

> > > ### Comment · Reviewer_9Zfz · 2025-11-25
> > >
> > > The part on "Validation with Human Experts" is quite important in my opinion, I'll raise my overall score given the responses, and I'd recommend highlighting this part somewhere in the main paper.

---

> > > > ### Author Response · Authors · 2025-11-25
> > > >
> > > > We sincerely thank the reviewer for the positive re-evaluation and the decision to raise the score.
> > > >
> > > > We fully agree with your insight that the **"Validation with Human Experts"** is a cornerstone of our work's validity. As per your strong recommendation, we will ensure this part is **prominently highlighted in the main text** in the final version (e.g., by moving key protocol details and inter-rater statistics from the appendix to Section 3 and emphasizing the 90.95% agreement in the abstract/intro) to ensure it receives the visibility it deserves.
> > > >
> > > > Thank you again for your time and constructive feedback, which has significantly helped us strengthen the paper.

---

### Author Response · Authors · 2025-11-19
**General Response**

We thank all reviewers for their time and thoughtful feedback. We have incorporated the rebuttal content into a revised version of the paper; all new or clarified text is highlighted in red in the manuscript. Below we briefly restate the main aim of **ArtifactsBench** and summarize the key improvements made during the rebuttal.

ArtifactsBench targets a setting that is increasingly important in practice: evaluating LLMs on **instruction-to-executable interactive visual artifacts**, including games, dashboards, management systems, SVG posters, and quick tools. Beyond static DOM or screenshot matching, our benchmark couples 1,825 executable tasks across 9 domains with a checklist-guided, multimodal evaluation pipeline that inspects code, visuals, and interaction via sandboxed execution, three staged screenshots, and a dual-referee MLLM-as-Judge setup. This design yields scores that are both fine-grained (10 vision/code dimensions, difficulty and interaction tiers) and strongly aligned with human judgment (up to 90.95% Pair ACC with experts and 94.4% rank consistency with WebDev Arena).

During the rebuttal phase, we have made the following substantive updates, which we hope will make the contribution and evaluation protocol clearer and more reproducible:

- Clearer task space and interaction coverage.
The main text now explicitly describes the range of interaction types (time-driven dynamics, event-driven UI, forms and validation, sliders/drag/zoom, multimedia, SVG/Canvas interactions, multi-step workflows, game mechanics) and reports the distribution over interaction levels: 396 / 117 / 536 / 776 tasks for Static Visual / Mild Dynamics / High Dynamics / Intensive Interactive. We also clarify that 1,429 tasks require non-trivial interaction beyond a single static view.

- More detailed and auditable data pipeline.
The dataset section and a new appendix subsection (“Detailed Introduction to Data Collection and Cleaning”) provide a step-by-step description of how tasks are sourced (expert showcases, course projects, blogs, code repositories, SVG and game examples, LLM visual-to-query), filtered, de-duplicated, rewritten, difficulty- and domain-tagged, assigned interaction levels, paired with 10-dimension checklists, and validated for solvability and ambiguity. We also give rough estimates of human effort for these stages.

- Additional evaluation studies and robustness checks.
We expand the human evaluation to report multi-rater agreement measures and score distributions, and we analyze the cost–accuracy trade-off of visual evidence (0 / 1 / 3 / 5 screenshots). New appendix sections study:
(i) Score reproducibility under repeated referee runs;
(ii) **Robustness to checklist phrasing and template variants**;
(iii) **Programmatic HTML/DOM quality checks**, showing that static DOM metrics are nearly saturated and weakly discriminative; and
(iv) Contamination detection using a prefix-completion protocol with ROUGE-L.
Together, these studies support that the scores are stable, robust, and not driven by superficial rubric wording.

- Clarified dual-referee design and practical reproducibility.
We explain more clearly that Gemini‑2.5‑Pro and Qwen2.5‑VL‑72B are used as **two independent referees**, each inducing its own leaderboard; we do not fuse their scores. New figures show highly consistent partial-order constraints between the referees and confirm version stability for Gemini‑2.5‑Pro. An additional appendix section lists the public API endpoints used for proprietary models, facilitating external reproduction of our baselines.

- Qualitative examples and sharper takeaways.
A new appendix section (“Visualization Results”) presents representative rendered cases from several domains (SVG posters, web applications, games, management systems, data-science dashboards) to help readers gauge benchmark difficulty and diversity. The analysis section has been sharpened to emphasize where models still struggle most (especially Intensive Interactive and complex management UIs) and to highlight the consistent advantage of instruction-tuned generalist models over coder-only or VL-only specialists, suggesting concrete directions for future model development.

We appreciate the reviewers’ feedback, which helped us strengthen both the clarity and the empirical foundation of ArtifactsBench, and we are happy to address any further questions.

---

### Author Response · Authors · 2025-11-29
**Summary of Rebuttal Updates and Score Improvements for Submission 4022**

Dear Area Chair,

Thank you very much for your time and for overseeing the review process of our submission.

We would like to respectfully bring to your attention that before the recent system issue occurred, we had already engaged in constructive discussions with all reviewers, successfully resolving their major concerns. Based on the timestamps and the content of their responses, the reviewers expressed clearly positive and encouraging feedback regarding our rebuttal, and **two reviewers explicitly stated that they had raised their scores**, resulting in an improved score profile of **8, 8, 6, 4** (from the initial 8, 6, 4, 4).

Specifically:

*   **Reviewer e67X (Score raised to 8):** Explicitly confirmed that their concerns were addressed, stating: *"I confirm that I can now access the dataset... Consequently, I increased the score to recommend acceptance."*
*   **Reviewer 9Zfz (Score raised):** Acknowledged the value of our additional human validation, stating: *"The part on 'Validation with Human Experts' is quite important in my opinion, I'll raise my overall score given the responses..."*
*   **Reviewer Wsco:** Confirmed that their primary concern regarding our baseline comparison was resolved: *"Thanks for further clarification; it resolves my concern around WebDev Arena."*
*   **Reviewer ANuP (Score maintained at 8):** Remained positive and supportive of the paper.

The reviewers collectively acknowledged the novelty of ArtifactsBench in bridging the evaluation gap for interactive visual artifacts and praised the rigor of our data pipeline and dual-referee protocol.

However, due to the recent system issue that may have reverted evaluations to earlier states, this updated feedback and the final scores (8, 8, 6, 4) might not be fully reflected in the current interface. We wanted to clarify this situation to ensure our successful rebuttal and the reviewers' updated endorsements are accurately considered.

We sincerely appreciate your understanding and your consideration of this matter.

Best regards,

The Authors of Submission 4022

---

### Meta-Review · Area_Chair_gACU · 2026-01-08

**Summary:**

The reviewers generally recognized the comprehensive nature of the benchmark and the effort involved in data construction. However, they raised significant concerns regarding the scientific novelty and the depth of the evaluation methodology:

Novelty and Contribution: Reviewers 9Zfz and Wsco questioned the novelty of the work compared to existing benchmarks like Design2Code, suggesting the contribution might be incremental. Reviewer Wsco specifically noted that benchmarking visual code generation is not a novel problem and that the "checklist" approach resembles prompt engineering rather than a new scientific methodology.

Evaluation Validity: Reviewers 9Zfz and Wsco initially questioned whether three screenshots fed to an LLM-judge are sufficient to evaluate dynamic interactions, suggesting this might miss long-horizon workflows or nuanced physics. Reviewer Wsco remained unconvinced that this approach is a rigorous enough proxy for true interactive quality, preferring agent-based evaluation.

Data and Process Transparency: Reviewers Wsco and e67X initially found the description of the data collection pipeline and human effort insufficient. Reviewer 9Zfz noted a lack of concrete examples in the initial submission.

Human Baseline: Reviewer Wsco questioned the validity of WebDev Arena as a gold standard, concerned it might just be a "vibe check" rather than a functional assessment.

**Reviewer Concerns:**

The authors addressed concerns regarding the lack of examples and data collection details in the rebuttal by expanding the appendix and providing visualization results. The authors clarified the nature of the WebDev Arena data, for which Reviewer Wsco seems happy with. The authors highlighted their 280-instance expert study with high agreement (90.95%), which Reviewer 9Zfz cited as a key reason for raising their score (although the score 4 is still on the negative side).

There are a number of outstanding issues. Despite the rebuttal, Reviewer Wsco retained the concern that the automatic evaluation mechanism (LLM-as-a-judge with a checklist) is essentially "prompt engineering" rather than a novel scientific contribution. Reviewer Wsco remained unconvinced that the current screenshot-based signal is a future-proof or scientifically rigorous method for evaluating interactive artifacts, suggesting that as models improve, static "looks" will yield diminishing returns as a signal for functional quality.

**Reviewer Scores:**

Reviewer ANuP: This reviewer maintained a score of 8 (Accept).

Reviewer e67X: Increased the score to 6

Reviewer 9Zfz: The reviewer promised to raise the score. It's shown as 4 in the system.

Reviewer Wsco: Despite acknowledging some clarifications, they maintained their score of 4 (Reject), arguing that the fundamental methodology lacks sufficient novelty and robustness.

In sum, although the engineering effort required to curate these tasks, implement the execution sandbox, and calibrate the checklist-based judge is evident and acknowledged by all reviewers, there lack significant scientific contributions over purely engineering feats. While the resource is comprehensive, the core methodology—using an MLLM as a judge with screenshots and checklists—is an application of existing techniques rather than a novel scientific advancement. As noted by Reviewer Wsco, the evaluation framework largely amounts to sophisticated prompt engineering. Furthermore, relying on static snapshots to evaluate dynamic interactions, while practically useful, does not fundamentally advance our scientific understanding of how to evaluate interactive systems in a way that agent-based or formal verification methods might.

---

### Decision · Program_Chairs · 2026-01-26

Reject